# Chiral emergence in multistep hierarchical assembly of achiral conjugated polymers

Kyung Sun Park [1], Zhengyuan Xue[1], Bijal B. Patel [1], Hyosung An [2], Justin J. Kwok[2], Prapti Kafle[1], Qian Chen [2], Diwakar Shukla[1] & Ying Diao [1,2,3,4,5 ✉]

Intimately connected to the rule of life, chirality remains a long-time fascination in biology, chemistry, physics and materials science. Chiral structures, e.g., nucleic acid and cholesteric phase developed from chiral molecules are common in nature and synthetic soft materials. While it was recently discovered that achiral but bent-core mesogens can also form chiral helices, the assembly of chiral microstructures from achiral polymers has rarely been explored. Here, we reveal chiral emergence from achiral conjugated polymers, in which hierarchical helical structures are developed through a multistep assembly pathway. Upon increasing concentration beyond a threshold volume fraction, dispersed polymer nanofibers form lyotropic liquid crystalline (LC) mesophases with complex, chiral morphologies. Combining imaging, X-ray and spectroscopy techniques with molecular simulations, we demonstrate that this structural evolution arises from torsional polymer molecules which induce multiscale helical assembly, progressing from nano- to micron scale helical structures as the solution concentration increases. This study unveils a previously unknown complex state of matter for conjugated polymers that can pave way to a field of chiral (opto)electronics. We anticipate that hierarchical chiral helical structures can profoundly impact how conjugated polymers interact with light, transport charges, and transduce signals from biomolecular interactions and even give rise to properties unimagined before.

---

[1] Department of Chemical and Biomolecular Engineering, University of Illinois at Urbana-Champaign, 600 S. Mathews Ave., Urbana, IL 61801, USA. [2] Department of Materials Science and Engineering, University of Illinois at Urbana-Champaign, 1304 W. Green St., Urbana, IL 61801, USA. [3] Beckman Institute, Molecular Science and Engineering, University of Illinois at Urbana-Champaign, 405 N. Mathews Ave., Urbana, IL 61801, USA. [4] Department of Chemistry, University of Illinois at Urbana-Champaign, 505 S. Mathews Ave., Urbana, IL 61801, USA. [5] Materials Research Laboratory, The Grainger College of Engineering, University of Illinois at Urbana-Champaign, 104 S. Goodwin Ave., Urbana, IL 61801, USA. ✉email: yingdiao@illinois.edu

Hierarchical structures are inherent to various soft material systems, including biomolecules, mesogens and conjugated polymers. Structural chirality commonly results from the hierarchical organization of chiral building blocks, e.g., amyloids, M13 phage and chiral mesogens[1–3]. In fact, chirality is common in nature and plays an essential role in various research fields such as biology, medicine, chemistry, physics and materials science[4–6]. For example, many biological substances including carbohydrates, amino acids and nucleic acids are chiral and their functional structures selectively respond to a particular chirality. Thus, understanding and controlling chirality across various length scales are important for developing functional soft materials. Recently, it has been discovered that achiral mesogens or metal oxide nanocubes can also form chiral, twisted structures due to symmetry breaking or topological defects[7,8]. In particular, achiral bent shaped molecules such as rigid bent-core mesogens and dimers linked with flexible chains have exhibited chiral/helical assembled structures, e.g., helical nanofilament phases and twist-bend nematic phases[9–11]. Also, it was observed that a rigid rod liquid crystalline polyelectrolyte[12], poly(2,2′-disulfonyl-4,4′-benzidine terephthalamide) (PBDT) forms the double helical macromolecule driven by numerous intermolecular interactions[13]. The synergistic array of various interchain interactions such as hydrogen bonding, dipole–dipole, and/or ion–dipole interactions allows PBDT to have the double-helical conformation and extreme rigidity with an high axial persistence length (~1 μm).

When applied to semiconducting and conducting polymers, such achiral-to-chiral transitions open a degree of freedom for tuning electrical, optical, biological and mechanical properties and can provide further fundamental understanding of complex supramolecular assembly and phase transition behaviors that occur during device fabrication. Conjugated polymers underpin a broad range of emerging technologies, due to their ability to transport charges[14,15], couple light absorption with charge generation[14–16], couple ion transport with electron transport[17], and transduce various interactions and stresses into electrical signals[18,19]. Thanks to such versatile functional properties, conjugated polymers have found use in electronics[20], thermoelectrics[21], solar cells[16], photocatalysts[22], electrochemical devices (fuel cells[23], batteries[24], supercapacitors[25]) and biomedical devices[19,22]. All the fundamental physical processes described above sensitively depend on multiscale morphology that includes polymer conformation, packing, crystallinity, alignment and domain connectivity[26]. Such complex morphology is, in turn, highly sensitive to molecular assembly pathways[27,28]. Particularly, recent studies have demonstrated that certain donor–acceptor (D–A) conjugated polymers readily form aggregates and/or lyotropic liquid crystalline (LC) phases in appropriate solvent systems[29–35]. Such intermediate states have shown benefits of their molecular assembly in terms of increasing crystalline domain size, inducing alignment, and thus enhancing device performance of thin films. While the lyotropic LC phases of conjugated polymers have been observed, little is known of their structures and hierarchical assembly, let alone supramolecular chirality. Lack of critical structural information severely hinders our understanding of how LC phases of conjugated polymers impact thin-film morphology and functional properties. Such knowledge on structure and in particular chiral emergence can further lead to optical, electronic, and mechanical properties unimagined before.

Herein, we report chiral emergence from achiral D–A conjugated polymers. We discover four liquid crystalline mesophases previously unknown to conjugated polymers, out of which three are chiral. We unveil a surprisingly complex hierarchical helical structures from the molecular, nanoscopic to micron scale. This insight is obtained by combining optical and electron microscopy imaging, optical spectroscopy, and X-ray scattering with molecular dynamic (MD) simulations. Our findings are significant as the states of matter discovered can redefine how we understand their optical, electronic, mechanical and biological properties of conjugated polymers which underpin a wide range of emerging technologies. This work also contributes to fundamental understanding of liquid crystals as this reports polymeric twist-bent mesophases whose formation mechanism is distinct from bent-core molecules and colloids.

## Results and discussion

**Polymer system and characterization.** The D–A copolymer used in this study is an isoindigo-bithiophene-based copolymer (PII-2T) (Fig. 1a) which is slightly twisted caused by the systematic torsion within each repeat unit[35]. A detailed molecular structure in the solution state was further explored using MD simulations, discussed later. Such isoindigo-bithiophene-based copolymers belong to a family of high performance p-type organic semiconductors, extensively studied over the past 10 years as active materials for transistors, solar cells, bioimaging, etc[36]. However, no thermotropic nor lyotropic liquid crystals have been reported for isoindigo based conjugated polymers so far. We surprisingly discovered several LC mesophases through cross-polarized optical microscopy (CPOM) and circular dichroism spectroscopy (CD). We then determined their nano- to micron-scale structures via electron microscopy techniques which revealed that the fundamental building blocks of the observed LC mesophases are polymer nanofibrils. The internal structures of the polymer fibrils were further analyzed via small-angle X-ray scattering (SAXS), grazing incidence wide-angle X-ray scattering (GIWAXS) and UV-Vis absorption spectroscopy. This wide range of characterizations together with MD simulation leads us to depict on how complex hierarchical helical structures emerge from achiral conjugated polymers. These results are discussed in detail below.

**Discovery and identification of LC mesophases.** We first observed emergence of several liquid crystal phases from isotropic solution during evaporative assembly of PII-2T in a receding meniscus (Supplementary Movie 1). Resembling the solution printing process, polymer solution traverses the entire concentration range from that of isotropic solution to that of solid thin film in the meniscus region. With increasing solution concentration, we observed nucleation of liquid crystals in confined droplets, merging of droplets into a continuous liquid crystal phase of uniform texture, followed by its transition to striped texture. To investigate the structure of each phase in depth, a series of PII-2T solutions at defined concentrations was prepared by a drop-and-dry method from chlorobenzene solution. By this approach, successive drop-casting from a stock solution (10 mg/ml) was performed between two glass slides to concentrate solutions to above 200 mg/ml. The sandwiched solution was annealed over multiple thermal cycles and equilibrated at room temperature to ensure reaching equilibrium state (see Materials and Methods). We note that, while chlorobenzene was used as the solvent, the same liquid crystal phases were observed in chloroform solutions as well. The morphology and optical properties of the PII-2T solutions are summarized in Fig. 1b, showing clearly that the mesophases are highly dependent on solution concentration, indicating a lyotropic LC nature. The solution prepared up to ~40 mg/ml displays no birefringence or apparent aggregates under CPOM. At ~50 mg/ml it begins to produce spindle-shaped birefringent microdroplets (tactoids), indicating a transition state where LC mesophases nucleate and grow from an isotropic phase. Detailed examination reveals that the mesophase at 50 mg/mL are, in fact, homogeneous nematic tactoids in which the director field, i.e., an average direction of the backbones is

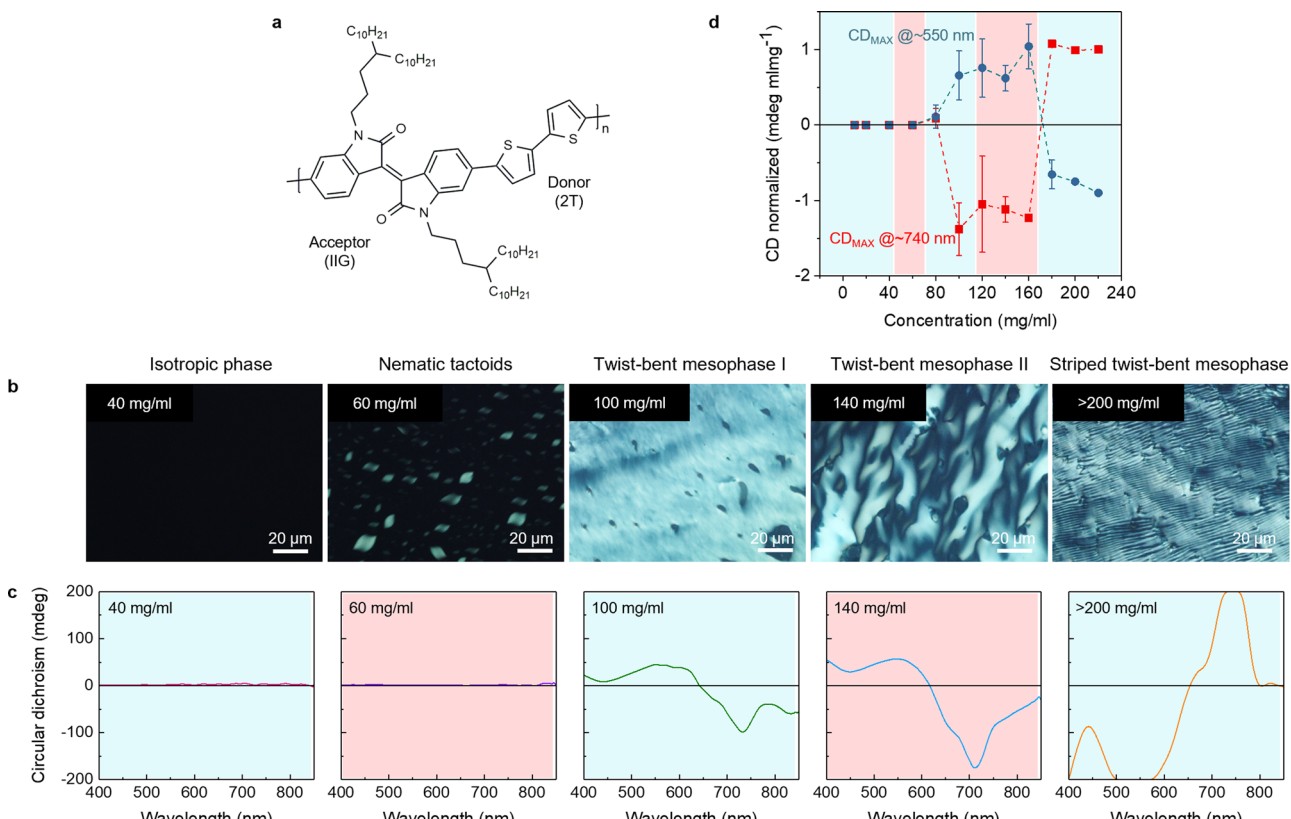

**Fig. 1 Chiral emergence of lyotropic LC PII-2T. a** Molecular structure of isoindigo-bithiophene–based copolymer (PII-2T). **b** CPOM images of the PII-2T solutions using transmitted light, showing various morphology of the LC mesophases. **c** CD spectra of the corresponding PII-2T solutions shown in **c**, indicating chirality emerges when exceeding a critical concentration (>60 mg/ml). **d** Averaged maximum values of concentration normalized CD for various solution concentrations where the values are obtained from two main peaks at ~740 nm (red dots) and ~550 nm (blue dots), respectively. The error bars are standard deviations. **c, d** The color code in each panel indicates the five regimes that we observed the distinct morphologies; (i) isotropic phase, (ii) nematic tactoids, (iii) twist-bent mesophase I, (iv) twist-bent mesophase II and (v) striped twist-bent mesophase (a detailed discussion later).

aligned to the long axis of the tactoids (Supplementary Fig. 1). As the solution concentration slightly increases to ~60 mg/ml, these homogeneous tactoids transition to bipolar tactoids, where the director field traces the curving edge of tactoids instead (Supplementary Fig. 2). The observed homogeneous to bipolar tactoid transition mirrors that of LC phases of amyloid fibrils when a specific critical volume is reached[1]. The tactoids observed in our study seem nonchiral as they show a mirror-symmetric birefringence with a dark center when the director is aligned parallel to either the polarizer or the analyzer (see Supplementary Figs. 1, 2), in contrast to optically active centers in the case of chiral tactoids[37]. This is because the twisted director in chiral tactoids forms a titled angle along the main axis of the tactoids. We note that this is only valid when the director is one handedness with a micron scale systematic twist. Further analysis of these nonchiral tactoids was carried out using CD discussed later. As the solution concentration further increases (~100 mg/ml), a uniform birefringent texture was observed, where discrete domains of isotropic phases (negative tactoids) are embedded in a continuous domain of mesophase. The solution at ~140 mg/ml showed complex morphology with micron-scale domains. At an extremely high concentration of ~200 mg/ml, a striped texture with a few micron periodicity was obtained, which resembles the zigzag twinned morphology of printed PII-2T solid films[35]. By rotating the sample under fixed crossed polarizers, the alternating dark and bright band patterns indicate the characteristic of twinned domains. Hereafter, we refer to each phase shown in Fig. 1b as (i) isotropic phase, (ii) nematic tactoids, (iii) twist-bent mesophase I, (iv) twist-

bent mesophase II and (v) striped twist-bent mesophase that are proposed based on comprehensive structural characterizations, discussed later.

We next used CD spectroscopy to explore the chirality of the observed mesophases. Figure 1c shows the CD spectra of the PII-2T solutions at solution concentrations corresponding to the CPOM images shown in Fig. 1b. The crystalline mesophases made at 100 mg/ml and above show very intense bisignate CD bands near 630 nm which corresponds to the main absorption wavelength of the polymer solution. This bisignate CD band represents chiral exciton coupling via Davydov splitting, indicating the polymer backbones are formed in a chiral fashion[38]. The mesophases at 100 and 140 mg/ml show negative and positive CD signals around 740 nm and 550 nm, respectively, indicating that the backbones form left-handed helical aggregation. The mesophase at >200 mg/ml shows inversed CD signs at each wavelength, indicating right-handed helical aggregation. We note that the handedness of each chiral mesophase exhibits a stochastic character. For a total fifty samples of each mesophase, left-handed twist-bent mesophase I and II form at 56% and 62% probability respectively, and right-handed striped twist-bent mesophase form at 78% probability. The preferential handedness inverts from twist-bent mesophase II to striped twist-bent mesophase. The origin of this handedness inversion is currently unknown, but some indication may be provided by comparison with natural cholesteric phases, where similar phenomena is often observed when the volume fraction (concentration) is high[39–41]. In these systems, the handedness inversion occurred when

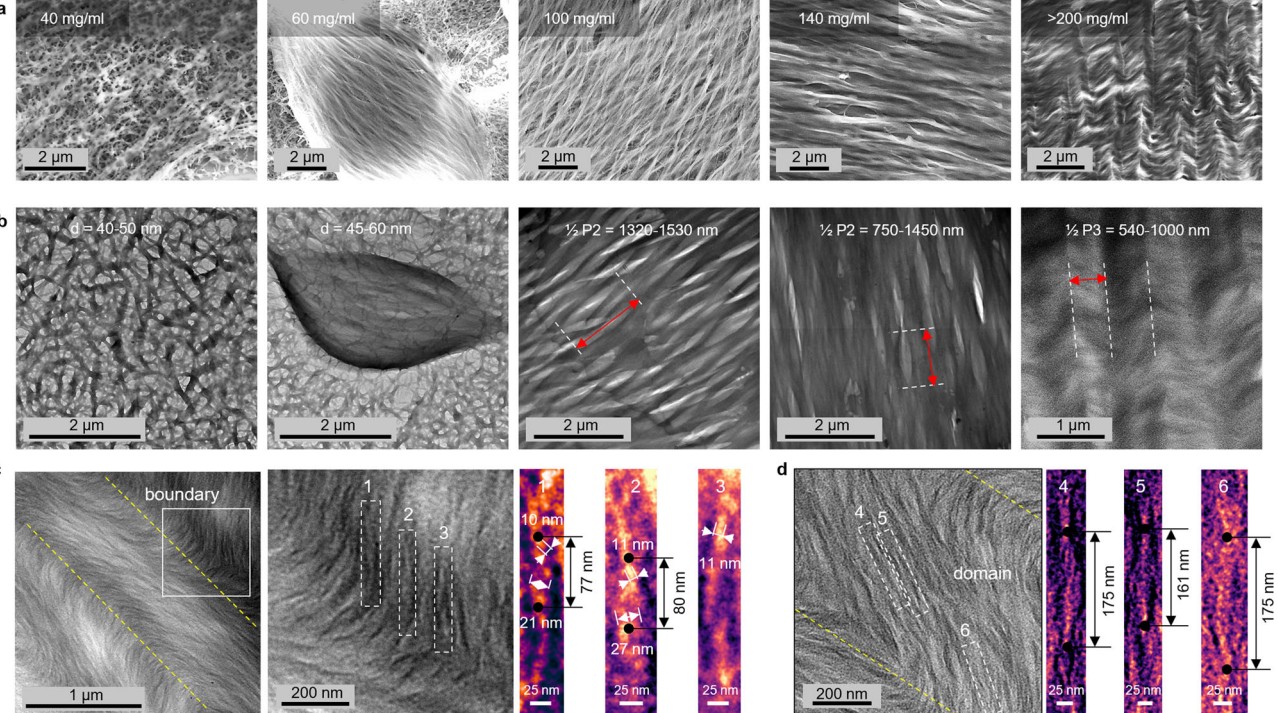

**Fig. 2 Micron and nanoscale morphology of PII-2T mesophase.** SEM (**a** top row) and TEM (**b** middle row) images of freeze-dried PII-2T mesophase at increasing solution concentrations (left to right). The half-pitch length marked with red arrows in TEM images exhibits a decreasing trend as concentration increases. High-resolution TEM images (**c**, **d** bottom row) of the printed PII-2T film, showing nanoscale twisted fibers on the domain boundaries and within the domains of the twinned thin film. The yellow dash lines denote domain boundaries. 1–6 notes selected regions showing twisted nanoscale fibers. Further imaging analysis was performed using ImageJ.

preferred packing structures resulted from a complex interplay of electrostatics, molecular sequences, excluded volume interactions, etc. Figure 1d shows a summary of the concentration normalized CD signals for various solution concentrations. In contrast to the twisted mesophases, the isotropic phase (<40 mg/ml) and the nematic tactoids (50–60 mg/ml) show zero or negligible CD signal. It is inferred that those phases are either nonchiral or form a near-racemic mixture. We speculate that the isotropic phase may exhibit both handedness equally (see discussions of MD results), while the nucleation and growth of mesophases drives the transition to single-handedness in a stochastic fashion.

**Mesoscale morphology characterizations.** Further electron microscopy imaging allowed us to explore in detail the nano- and micron-scale structures formed within the mesophase. SEM (Fig. 2a) and TEM images (Fig. 2b) were obtained from samples that were freeze-dried at concentrations corresponding to the entire series of achiral, isotropic to chiral, and twisted mesophases. We confirmed by CD that chirality of the samples was preserved in freeze-dried samples (Supplementary Fig. 3). In the isotropic phase (~40 mg/ml), polymer fibrils with a diameter of 40–50 nm was observed. The cross-section of fibers was revealed to be nearly circular after measuring the height of individually dispersed fibers (Supplementary Fig. 4). In tactoids (~60 mg/ml), the fibers bundle up to form spindle-shaped tactoids with the fiber long axis aligned with curving edge of the tactoid. At ~100 mg/ml, fiber bundles merge into a continuous domain of aligned fibers. The interesting feature in this phase is the locally twisted fibers with a half-pitch length of 1.3–1.5 μm (Supplementary Fig. 5). At ~140 mg/ml, the fibers become denser, thicker where the half-pitch length becomes shorter (0.7–1.4 μm). As the solution concentration further increases to around 200 mg/ml, a

zigzag twinned morphology is observed with a domain width of 0.5–1 μm. This twinned morphology may arise from concerted twisting and bending of densely packed polymer fibers. In addition to micron-scale helicity observed by SEM, high-resolution TEM imaging further unveiled nano-scale helicity of the zigzag twinned morphology (Fig. 2c, d). Due to the difficulty of capturing high-resolution TEM on relatively thick freeze-dried solution samples (~1 μm), printed thin films (~200 nm) that showed resembling morphology were employed instead. Intriguingly, we observed twisted nanoscale fibers (~10 nm) of the twinned morphology, with a helical pitch length of ~170 nm at the domain and with the pitch length of about 80 nm at the boundary. We note here that there should be at least three scales of helicity that span the molecular, nano to micron scales (shown later). We will show through molecular simulations that the nanoscale and micron scale helicity likely arise from the molecular scale helicity.

The electron microscopy imaging clearly reveals that the LC mesophases are comprised of dispersed polymer fibrils, rather than single polymer chains. In other words, we infer that mesophases are colloidal, not molecular LCs, validation of which requires directly probing the solidification process using in situ characterization techniques[42]. Further, the imaging result shows that it is when the fibers become helically twisted that the chirality emerged, at 100 mg/ml. This points to a possible link between chiral emergence and the helical structure of the fiber. With further increasing concentrations, the helically twisted fibers further assemble into higher-order structures that take the form of micron-scale twinned domains. Such hierarchical helical assembly of conjugated polymers is reminiscent of biological assembly of helical structures, such as chiral amyloid fibers[1] and chiral M13 phage particles[2]. The helical rod-like shaped M13 phage with a helical pitch length of 3.3 nm assembled into a

wavelike "ramen noodle" morphology in a dip-coated thin film. The film exhibited a supramolecular twist pitch length of ~10 μm. Translation of nanoscale helicity to micron-scale twists was attributed to both chiral liquid crystalline phase transitions and competing interfacial forces at the meniscus. Also, this periodic morphology has been observed for chiral nematic phase such as cholesteric phase[43,44], twist-bend nematic phase[45] or heliconical smectic phase[46]. Because there is clearly no molecular chiral center in the polymer and solvent molecules used in this work, the helical mesophases we observed is closer to the twist-bent nematic phase or heliconical smectic phase where the helical assembly originates from symmetry breaking of bent shaped small molecules or colloids. However, as we shall show that the molecular underpinnings of our helical mesophases is distinct from bent-core mesogens.

**Molecular scale structure characterizations.** Next, we determined the internal structure of the polymer fibers constituting the mesophases by characterizing the polymer conformation and intra-fiber molecular packing using small-angle X-ray scattering (SAXS) and grazing incidence wide-angle X-ray scattering (GIWAXS). SAXS samples were prepared by the injection of pristine-made solutions into glass capillaries. Figure 3a shows the solution SAXS analysis at various solution concentrations. Note that the SAXS analysis of the solution around 200 mg/ml is absent due to the extremely high viscosity that hinders capillary filling. All solutions show a peak around 0.19–0.22 Å$^{-1}$ which corresponds to lamellar stacking within the polymer nanofibers. This result indicates that the fibers observed from the EM imaging analysis are semicrystalline. Further deconvolution and data fitting provide the lamellar stacking distance inside the fibers and the radius of single polymer chains. The values were obtained through 1D SAXS fitting algorithms reported in our recent work[47]. Briefly, both polymer fibrils and dispersed polymer chains co-exist in chlorobenzene solutions with the lamellar peak resulting from lamellar stacking of polymer chains within the fibrils. Therefore, our model consists of a power law, semiflexible cylinder, and pseudo-Voigt peak to fit the contributions from the polymer fibrils, dispersed polymer chains, and lamellar peak, respectively. In all samples, a low q power law exponent of around −3.4 was observed, indicating Porod scattering from large fibrils which is in agreement with our observation that polymer fibrils constitute the mesophases. The semiflexible cylinder and pseudo-Voigt peak were used to extract the radius of the polymer cross-section and the lamellar stacking distance, respectively. Figure 3b shows the lamellar stacking distance increases as solution concentration increases. To explain this, we propose that as polymer chains become more twisted, the effective volume of polymer chains may be increased and thus molecular packing in the fibers may be loosened, leading to increased lamella stacking distance. This observation is in line with observations reported in previous studies that helical structures lead to increased interchain spacing[48,49]. The radius of the polymer chain cross-section (Fig. 3c) closely matches with half lamellar distance in most cases, indicating absence of side-chain interdigitation in all the solution phases. We note that typical lamellar stacking distance of solidstate films is ~25 Å, which is smaller than the values (28–33 Å) we obtained from solution phases. We attribute larger lamella stacking distance in solution phases to more disorder and/or more expanded/swollen side-chain conformation due to the presence of solvent.

We then utilized GIWAXS to characterize molecular stacking and orientation distribution within these mesophases. Freeze-dried solution samples used for imaging analysis were investigated through GIWAXS. Two-dimensional (2D) GIWAXS

patterns of the mesophases are provided in Fig. 3d. We observed the (010) π–π stacking and (100) lamellar stacking peaks for all samples, which affirms the semi-crystallinity of polymer fibers. Molecular packing details for the π–π and lamellar stacking peaks are summarized in Supplementary Table 2. According to the MD simulation result (see the MD simulation section) and our previous study[35], we infer that the π-stacking distance in solution-dispersed polymer fibrils is ~4.4 Å. This diffraction signature overlaps with the amorphous ring in GIWAXS, so it is very challenging to differentiate from one another. The π-stacking distance at ~3.6 Å obtained from GIWAXS may arise from crystallization of dispersed polymer fibers during the freeze-drying process. Therefore, we focus our analysis on the lamella stacking peak below. Pole figure analysis[50] performed on the lamellar stacking peak as a function of the polar angle (χ) provides indirect evidence of twisted nanofibers in the mesophases through molecular orientation distributions (Supplementary Fig. 6). A distinct preferential edge-on orientation of lamellar stacks is observed in the isotropic phase (10 mg/ml) with a minor contribution from face-on stacks (Fig. 3e, left). The degree of out-of-plane orientation was quantified using the 2D orientation parameter, $S_{2D} = 2\langle\cos^2 \gamma\rangle^{-1}$, where $\gamma$ is the tilt angle of the lamellar stacking direction with respect to the substrate (Fig. 3e, middle). Values of $S_{2D} = 1$, 0, and −1 correspond to perfect face-on, isotropic and perfect edge-on orientation, respectively. With the appearance of mesophases at and beyond ~60 mg/ml, the lamellar stacking orientation becomes more isotropic (Fig. 3e, right), which we believe is associated with twisted polymer backbone in helical mesophases.

We further inferred molecular conformations of these mesophases by UV-Vis absorption spectroscopy. According to our linear POM and polarized UV-Vis spectroscopy (Supplementary Figs. 7, 8), the polymer chains are aligned along the fiber-long axis with the optical transition dipole moment occurring mainly along the polymer backbones. Figure 4a shows UV-Vis absorption spectra of PII-2T solutions that correspond to the series of isotropic to twisted mesophases. There are several main peaks positioned around 715 nm, 650 nm, 420 nm, and broaden peaks around 500–600 nm. First of all, the two peaks around 715 nm and 650 nm clearly indicate vibronic character of the same excitonic transition given the difference of 1400 cm$^{-1}$ (0.17 eV) between those two peaks; this wavenumber (energy) corresponds to the vibrational frequency of the aromatic-quinoidal stretching mode for nearly all π-conjugated molecules[51]. Therefore, the peak at 715 nm can be assigned as the ground-state electronic transition (0–0) and the 650 nm peak its higher order vibronic transition (0–1). The far higher energy transition around 400 nm is attributed to localized transition from isoindigo units[35] which becomes dominant when the polymer chains are more torsional and the electrons are more localized. The relative intensity of the vibronic progression characterized by the absorption peak ratio (0–0)/(0–1) has been used to indicate conjugation length and polymer conformation[51–53]. Specifically, the decrease in absorbance ratio (0–0)/(0–1) as well as the sharp decline of absorption coefficient with increasing solution concentration suggest a decrease in π-conjugation and increase in backbone torsion from isotropic to twisted mesophase (Fig. 4b). Moreover, the (0–0) and (0–1) peaks are blue-shifted with increasing solution concentration (Fig. 4c), showing that the transition energy increases owing to decreasing conjugation length caused by torsional polymer chains.

**Molecular simulations unveil helical conformation.** We next performed MD simulations on PII-2T to answer how structural features at the molecular level may lead to chiral emergence. We

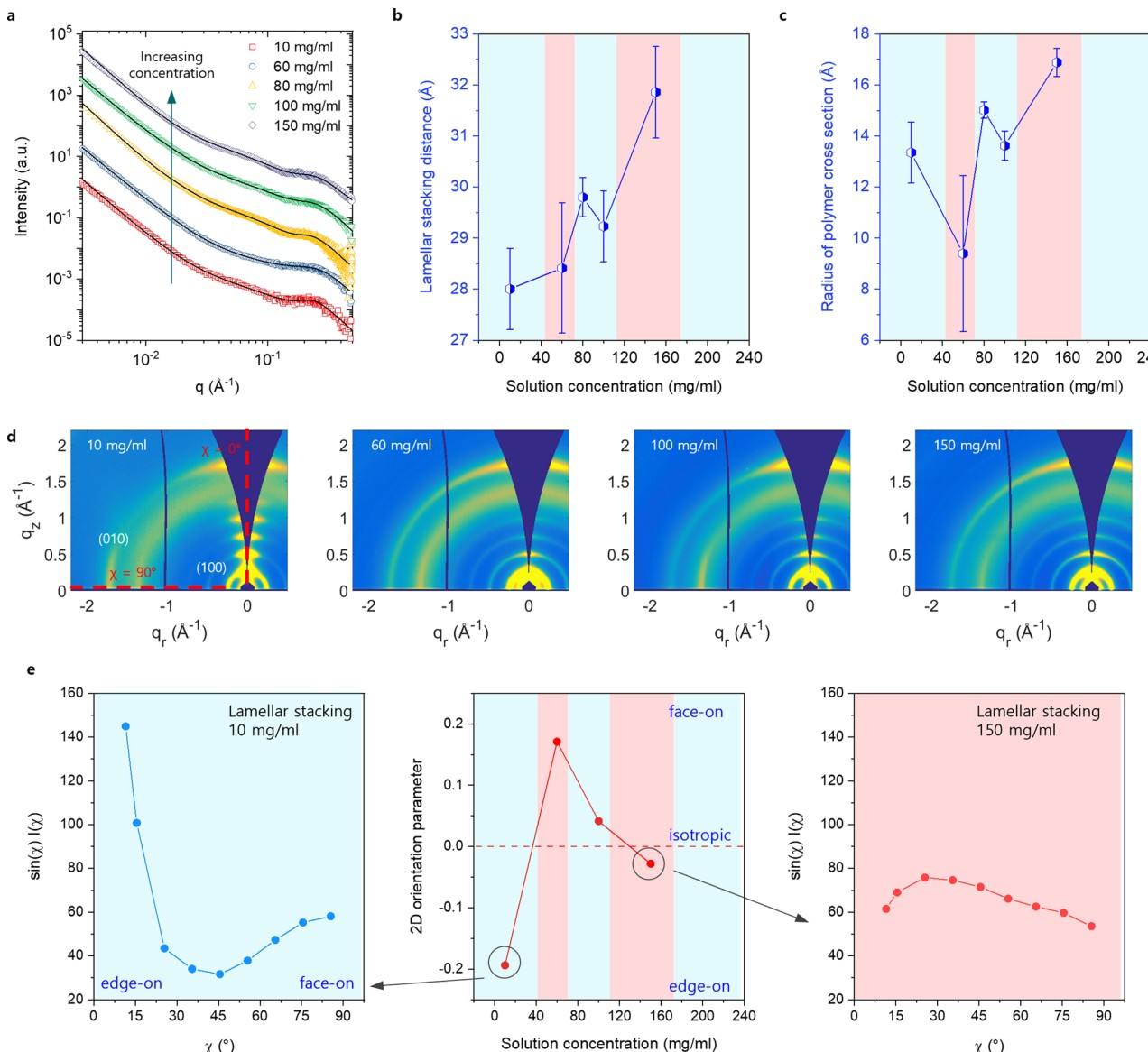

**Fig. 3 Molecular scale assembly of PII-2T mesophase. a** SAXS plots of PII-2T mesophase with increasing solution concentration (as labeled). Black solid lines are fitting results using the model developed in ref. [44]. **b** Lamellar stacking distance obtained from the fitting, showing an increasing trend as solution concentration increases. The error bars are obtained from fitting. **c** Radius of the cross-section of a single polymer chain extracted from the Guinier knee near 0.07 Å$^{-1}$. The error bars are obtained from fitting. **d** 2D X-ray scattering patterns for the freeze-dried mesophases prepared at the indicated concentration. **e** Out-of-plane 2D orientation parameter of the mesophases (middle). $S_{2D}$ value closer to −1 or 1 indicates an edge-on and face-on orientation, respectively and $S_{2D} = 0$ indicates an isotropic orientation. Geometrically corrected intensity (**I**) of lamellar stacking (100) peak as a function of polar angle, $\chi$ for isotropic phase at 10 mg/ml (left) and twisted mesophase at 150 mg/ml (right). Note that $\chi = 0°$ and 90° correspond to edge-on and face-on orientation, respectively. **b**, **c**, **e** The blue and red color codes in each panel indicate the five distinct solution phases outlined in Fig. 1.

explore two possible molecular origins: first, polymer chains may adopt wavy, helical conformation arisen from flexible yet highly coupled dihedrals; second, molecular stacking may occur in a staggered fashion leading to chiral assembly. The latter point was shown in three other systems, where chiral aromatic peptide[54], chiral nanoplatelets[55] and achiral nanocubes[8] were found to stack at an angle giving rise to chiral helical structures.

To determine polymer conformation in solution, we constructed a system containing a 30-mer of PII-2T surrounded by chloroform and simulated it without any bias for ~260 ns. The conformation of the entire polymer chain fluctuates with multiple regions exhibiting local helical structure (Fig. 5a; Supplementary Movie 2). The half-pitch of the helix is comprised of 5 to 6 monomers corresponding to a length of about 60–90 Å. To

understand the molecular underpinnings of this intriguing phenomenon, we closely examined the distribution of four dihedral angles at various positions along the 30-mer (Supplementary Fig. 9, 10). The four dihedral angles are illustrated in Fig. 5b: the angle between the thiophene–isoindigo (T–I), the internal angle in the isoindigo unit (I–I), the angle between the isoindigo-thiophene (I–T) and the angle between the thiophene-thiophene (T–T). The I–I dihedral angle is the most rigid, with the narrowest distribution centered around 25° (Supplementary Fig. 9). The I–T and T–I dihedrals fluctuate between four well-defined positions (two cis and two trans conformations), all deviating from co-planarity by ~30°. In contrast, the T–T dihedral angle is the most flexible and fluctuates between +150° and −150°, with two peaks around ±90° (Fig. 5c). This

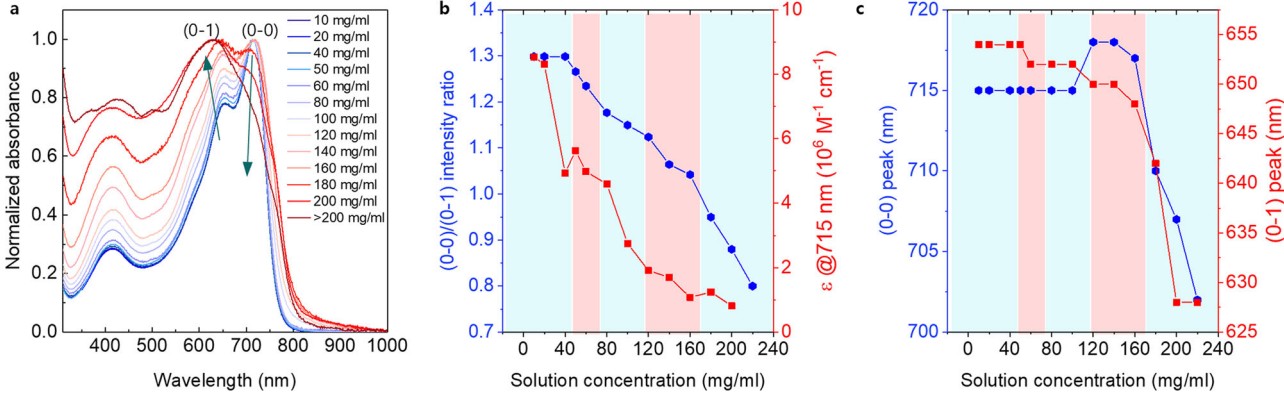

**Fig. 4 Optical transition property of PII-2T mesophase. a** UV-Vis absorption spectra of PII-2T solutions that corresponds an entire series of achiral, isotropic to chiral, twisted phases. The arrows indicate peak changes with change increasing solution concentration. **b** The decrease in both the absorption peak ratio, (0-0)/(0-1) and absorption coefficient (ε), indicating twisted molecular conformation as concentration increases. **c** Hypsochromic shift of the (0-0) and (0-1) peaks, showing that the dispersed polymer fibrils are more twisted and/or polymer conformation is more torsional as well. The blue and red color codes in **b**, **c** indicates the five distinct phases corresponding the morphology transition observed above.

suggests that the T–T dihedrals may act as flexible hinges, enabling the formation of wavy polymer conformation. Indeed, we observed a small but discernable preference towards cis conformation of T–T dihedrals in curved regions along the polymer chains, where two successive thiophene rings are oriented towards the same direction (Fig. 5d, Supplementary Fig. 11). In addition, we show that adjacent T–T dihedrals in helical regions are highly correlated (Pearson's $r > 0.7$) shown in Fig. 5e and Supplementary Fig. 12; this high extent of correlation further explains existence of helical conformation in solution dispersed single polymer chains.

To determine whether there is local chiral symmetry breaking at the single polymer level, we analyzed handedness of the simulated helices by counting ~3000 frames (Supplementary Fig. 13). We observed almost equal probability to form helix in both handedness (Fig. 5f). Therefore, we conclude that there is no intrinsic tendency for the isolated polymer to form helices with one dominant handedness. In sum, the MD simulations provide evidence that helical conformations with both handedness exist in solution at a single polymer level due to flexible, correlated dihedrals along the polymer backbone. However, there is absence of local symmetry breaking at the single polymer level. Nonetheless, weak asymmetric dihedral angle distributions do exist and persist over time (Supplementary Figs. 14, 15). While such weak asymmetry does not seem to significantly bias the handedness at the single polymer level, it could be amplified during multimolecular assembly process when adjacent polymers strongly interact. However, whether such dihedral asymmetry play a role in global symmetry breaking during chiral mesophase formation remains unknown.

To understand whether the fashion of molecular stacking contributes to structural chirality, we simulated dimeric assembly of PII-2T oligomers in chloroform. To reduce the computational cost, we substituted the long alkyl side-chains ($C_{24}H_{49}$) with short ones ($C_4H_9$) and constructed hexamers to better observe the interaction between backbones. During the ~400 ns simulation (Supplementary Movie 3), the hexamers that were 20 Å apart at the beginning took ~100 ns to form a dimer with their backbones aligned in a stable parallel conformation (Fig. 6a). The interaction between polymer backbones comes from the stacking between their thiophene and indole rings. To further explore the thermodynamics of interaction between backbone aromatic rings, we performed umbrella sampling simulations[56] to obtain the potential mean force (PMF) curve for dimerization between the aromatic rings. The center of mass

distance ($I$) and the dihedral angle ($\theta$) between the facing indole ring and the bithiophene rings of the two hexamers were used as two reaction coordinates (Fig. 6b). First, Umbrella sampling was performed varying $I$ from 3 to 13 Å using a window size of 0.5 Å. The PMF (Fig. 6c) showed that the free energy of dimerization reached the minimum at ~4.6 Å, which closely matches with the average π-stacking distance (~4.4 Å) in amorphous regions of the thin films measured by GIWAXS[35]. We also performed Umbrella sampling along $\theta$ and the PMF curve showed the free energy minimum at ~45° (Fig. 6d), which is comparable to the dihedral angle observed in the unbiased simulation. Interestingly, the potential well around the ~45° minimum was found to be asymmetric. Such staggered, asymmetric dimeric stacking may play a role in global symmetry breaking during chiral mesophase formation. However, we note the absence of symmetry breaking at the single polymer fibril level inferred from absence of chirality in nematic tactoids. In sum, we spotted in our simulation consecutive twist and bent oligomers that are dimerized in a helical fashion via the indole-thiophene stacking with a half-pitch length of ~50 Å (Fig. 6e). We believe the existence of such helical conformation and assembly forms the structural basis to chiral mesophase formation as discussed later.

**Hierarchical morphology and symmetry breaking mechanism.** All above characterizations culminate in a multiscale morphology model to reveal the chiral emergence of achiral conjugated polymers via a multistep hierarchical assembly process (Fig. 7). In dilute solutions, single polymer chains are prone to adopting wavy helical conformation owing to flexible, correlated dihedrals that easily accommodate curvature modulation via cis-trans transition. Both left- and right-handed chiral conformations exist transiently at the single-molecule level with almost equal probability (50.5% left-handed). Upon aggregation, polymer backbones π-stack in a staggered fashion forming helical nanofibers of 40–50 nm in diameter with a pitch length of 10–20 nm. We infer that both handedness exist at the single fiber level while the ensemble average over all fibers remains achiral. The nanofibers constitute the isotropic solution until nematic tactoids nucleate at ~50 mg/ml wherein nanofibers aggregate into spindles confined in mesophase droplets. The absence of global chirality may be explained by the symmetric shape of the spindles or the statistical distributions of chirality from droplet to droplet. Increasing concentration to 100 mg/ml

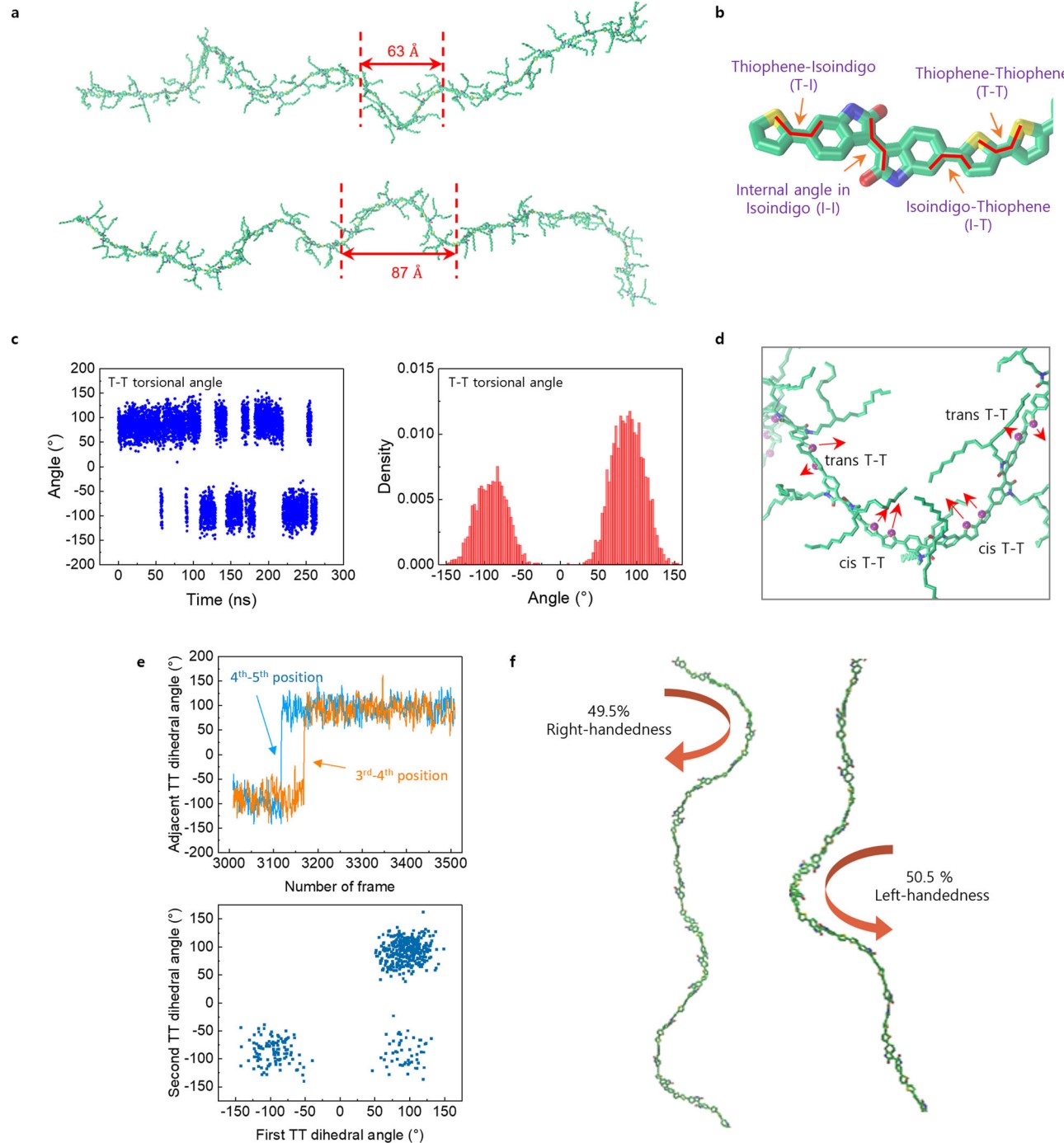

**Fig. 5 Helical conformation of PII-2T molecules in a solution phase. a** MD simulation of a 30-mer of PII-2T in chloroform, showing helix-like flexible structures with a half-pitch length of about 60–90 Å. **b** PII-2T molecular structure (omitted alkyl side chains and hydrogens) with each bond used in dihedral angle frequency plots. **c** T–T torsion angle plot between the 14th and 15th monomers of the 30-mer (left) and the corresponding angle distribution histogram (right). **d** Enlarged structure of the curved region on the helix structure. The purple dots are sulfur atoms on the thiophene rings. The arrow marks indicate a local conformation of the thiophene rings, showing a preferential cis-conformation in the curved region. **e** Selected time-dependent dihedral angle plots of the adjacent T–T pair (top) and the dihedral angle distribution of the adjacent T–T pair (bottom). **f** Fraction of frames that show either left- or right-handedness in total counts of frames showing helicity. Captured examples of PII-2T 30-mer showing a right- and left-handed helix formed during simulation.

eventually causes nematic tactoids to merge into a single coherent mesophase, twist-bent mesophase I, while leading to global chiral symmetry breaking. Helical, twisted fiber bundles constitute this mesophase, forming uniformly aligned and birefringent domains over large area; this suggests long-range interactions between fibers to induce single-handedness

arrangement, with left-handed chirality to appear at slightly higher probability (56%). Further increasing concentration densifies the fibers, thickens the fiber bundles and reduces the helical pitch in twist-bent mesophase II, while maintaining the preference towards left handedness as twist-bent mesophase I (at 62% probability). Ultimately, striped twist-bent mesophase

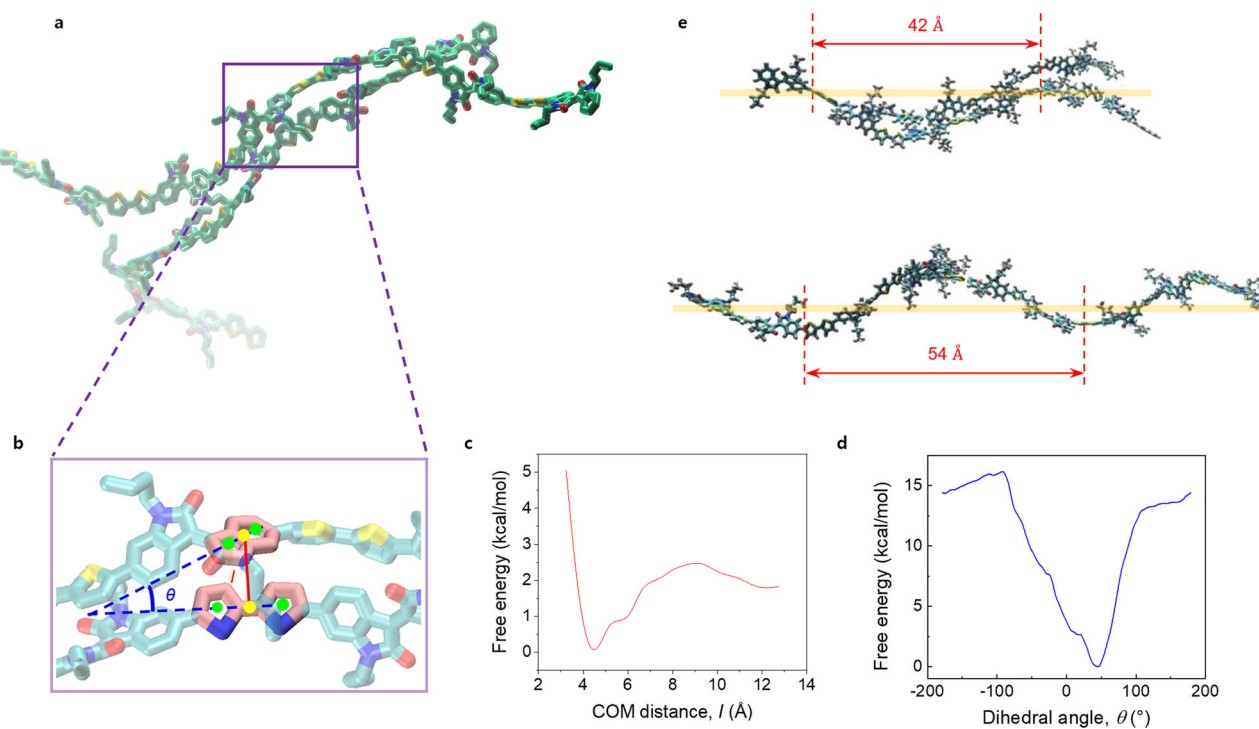

**Fig. 6 Helical structure of stacked PII-2T molecules in the solution phase. a** Dimeric assembly of PII-2T hexamers captured from the simulation, showing the aromatic rings stacked closely. **b** Reaction coordinates used for umbrella sampling. Red stick indicates the center of mass distance, $l$ between the bithiophene and indole rings (the center of mass denoted by yellow dots). Blue stick and dashed line indicate the dihedral angle, $\theta$ formed by the center of mass of these four rings (the center of mass denoted by green dots). **c** PMF plot as a function of the distance, $l$ obtained from umbrella sampling simulation using Weighted Histogram Analysis Method (WHAM), showing the minimum free energy at ⌣4.6 Å. **d** PMF plot as a function of the angle, $\theta$, showing the minimum free energy at ⌣45°. **e** Stacked oligomers captured from the simulation, showing the helical molecular assembly with a pitch length of about 50 Å.

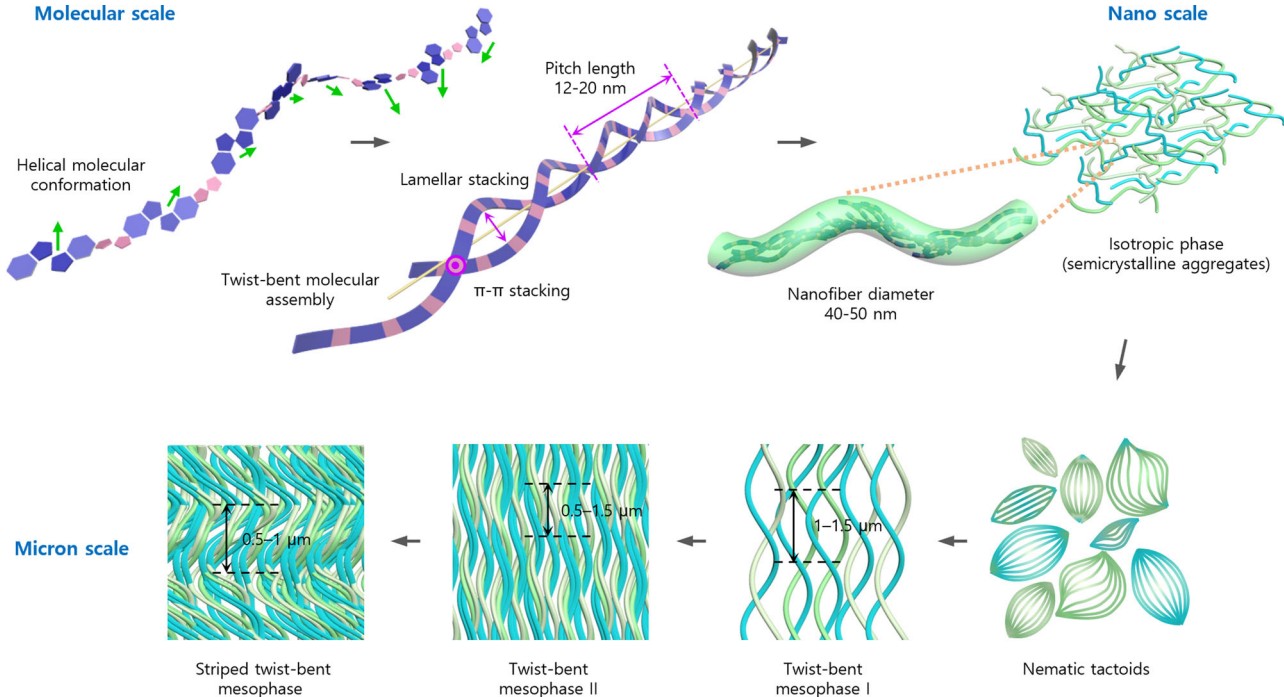

**Fig. 7 Schematic illustration of proposed chiral emergence in multistep hierarchical assembly of achiral conjugated polymers.** Wavy helical polymer chains are aggregated in a staggered fashion forming helical nanofibers. The nanofibers constitute the isotropic solution. Increasing concentration eventually leads nematic tactoids nucleate to merge into a coherent mesophase, twist-bent mesophase I. Further increasing concentration causes dense and thick fiber bundles and reduces the helical pitch length in twist-bent mesophase II. Ultimately, striped twist-bent mesophase emerges where densely packed fibers twist and bend coherently to result in micron scale zigzag twinned domains.

appears where densely packed fibers twist and bend coherently to result in micron scale zigzag twinned domains while reversing the dominant handedness of twist-bent mesophase II (frequency of occurrence for right handedness at 78%). Overall, increasing concentration enhances backbone torsion at the molecular scale, loosens lamella stacking at the mesoscale, and decreases helical pitch length at the microscale according to UV-Vis, GIWAXS, SAXS, and EM analysis. The structural changes at the molecular scale probably underlie the structural transitions observed.

What is the symmetry-breaking mechanism that underpins formation of chiral mesophases? We propose that handedness of the mesophase is stochastically "chosen" during the phase transition. In a racemic solution of chiral helical polymer fibrils, a population bias towards a certain handedness can transiently exist due to stochastic fluctuations. At the time when the polymer fibrils coalesce/nucleate into a chiral mesophase, such population bias can be amplified through conversion of the minority into the majority handedness, driven by intermolecular interactions depicted in Fig. 6 and free energy minimization when forming a coherent mesophase with uniform handedness. This view is supported by the observed stochastic nature of chiral symmetry breaking—the twist-bent mesophases I, II and striped twist-bent mesophase can all adopt both handedness with certain probabilities. It remains a question, however, whether the experimentally observed biases towards one handedness in a chiral mesophase relates to asymmetries at the molecular scale, namely asymmetric dihedral angle distribution and asymmetric backbone stacking observed in MD simulations. While such molecular asymmetries do not lead to symmetry breaking at the single polymer or single nanofibril level, nuanced imbalances at the molecular level could be amplified during the multimolecular assembly process. Indeed, we observed that bias towards certain handedness increases with increasing volume fraction/concentration of the polymer in solution. This suggests that asymmetric intermolecular interactions may play an important role in chiral symmetry breaking.

Is the observed emergence of chiral mesophases unique to PII-2T or more general among conjugated polymers? Interestingly, similar zigzag twinned morphology has been previously observed for several well-studied conjugated polymer systems, for instance, poly(3-hexylthiophene) (P3HT)[57], poly-2,5-bis(3-alkylthiophen-2-yl)thieno[3,2b]thiophenes (pBTTT)[58], naphthalene diimide-bithiophene-based copolymer (P(NDI2OD-T2))[29,34] and diketopyrrolopyrrole-benzothiadiazole-based copolymer (DPP-BTz)[35,59]. These polymers were also found to have a lyotropic LC nature. Zone-cast pBTTT has exhibited aligned micron scale domains with alternating backbone orientation tilted by about 45°[58]. Meniscus-guided coated P(NDI2OD-T2) and DPP-BTz films also showed the zigzag twinned morphology in which the wave-like micron scale structures formed along the coating direction[34,35,59]. Aged P3HT solution confined in a rectangular capillary has exhibited liquid crystalline orders with similar alternating dark and bright stripes that oriented perpendicular to the capillary-long axis[57]. While reported before, such zigzag morphology of conjugated polymer thin films has not been previously associated with chiral mesophases in previous reports, nor are the multiscale structure explored. On the other hand, this unique morphology seems to be strongly associated with structural chirality as similar zigzag twinned morphology was formed by chiral colloidal particles (M13 phage)[2] or bent-core mesogens[60]. Previous reports on similar film morphologies from multiple classic conjugated polymer systems suggest that our observed chiral emergence in lyotropic liquid crystal phases may be more general.

**Helical structure promotes charge generation**. Given that multiscale helical structures of achiral conjugated polymers are rarely observed before, we are interested in understanding how such structures impact optoelectronic properties. In particular, we performed photoluminescence spectroscopy (PL) to investigate how helicity impacts charge generation in bulk heterojunction (BHJ) organic solar cells (donor and acceptor structures and energy level alignment shown in Fig. 8a, b). We were able to control the helicity of the conjugated polymer films by tuning printing regimes as reported in our recent work[35]. Specifically, we prepared helical and non-helical blended films using PII-2T (donor) and PC71BM (acceptor) printed in the evaporation (0.005 mm/s) and transition (0.05 mm/s) regime, respectively. CD spectroscopy confirmed the chiral and nonchiral characteristics of the printed BHJ films (Supplementary Fig. 16). Steady-state and time-resolved PL were employed to compare charge generation properties in helical vs. non-helical systems (Fig. 8). The steady-state photoluminescence (PL) (Fig. 8c, d with the axis on the right) was about 52% decreased for the helical BHJ system whereas about 9% decreased for the non-helical BHJ system when compared to each neat polymer film. In the time-resolved PL (Fig. 8e, f), the decay time of radiative process (fluorescence) for the helical BHJ film is about 37% decreased whereas it is about 11% decreased for the non-helical BHJ system. A clear decrease in PL decay time of the helical BHJ system indicates an increase in energy transfer efficiency between the PII-2T donor polymer and the fullerene acceptor, in line with the steady-state spectral observation. We further observed a slower PL decay in the helical neat polymer film (0.332 ns) than the non-helical neat films (0.167 ns), suggesting a longer singlet exciton lifetime[61] in the helical film. This result shows the helical system could facilitate excitons in reaching the donor–acceptor interface for electron-hole dissociation. Together, steady-state and time-resolved PL results suggest that helical structure of donor polymers possibly leads to a longer exciton lifetime and more efficient charge splitting at the donor–acceptor interface compared to non-helical structures, which may be related to reduced energetic disorder upon helical assembly. The exact mechanism of this observation remains unclear and will be investigated in future works.

In summary, we report helical structures of achiral conjugated polymers that were developed through the multistep hierarchical assembly pathways. Through a combination of in-situ and ex-situ optical and electron microscopy, along with X-ray scattering, we observed that LC mesophases evolve from dispersed polymer nanofibers to assemble into chiral LC mesophases as the fibers become more twisted and bundle up in a helical fashion. Molecular dynamic simulations suggest that chiral helical conformation exists in solution at the single polymer level due to flexible, correlated dihedrals along the polymer backbone. Such single polymer chains further stack in a staggered fashion to form chiral helical fibrils. However, both handedness exist with almost equal probability at the single polymer and single fiber level. Informed by experimental observations, we propose that global symmetry breaking occurs in a stochastic fashion during chiral mesophase formation to reach a lower free energy state of having uniform handedness in a long-range ordered liquid crystal phase. Thus, by considering the role of achiral-to-chiral transition through hierarchical assembly, we are able to decipher the structural evolution of this conjugated polymer across concentration regimes spanning the dilute to thin-film states—an assembly pathway underlying all evaporative solution coating and printing processes. This study presents a multiscale picture of the structural origin of chiral emergence—a phenomenon general among multiple classes of conjugated polymers which has been overlooked before. We anticipate that the ability to precisely control chiral

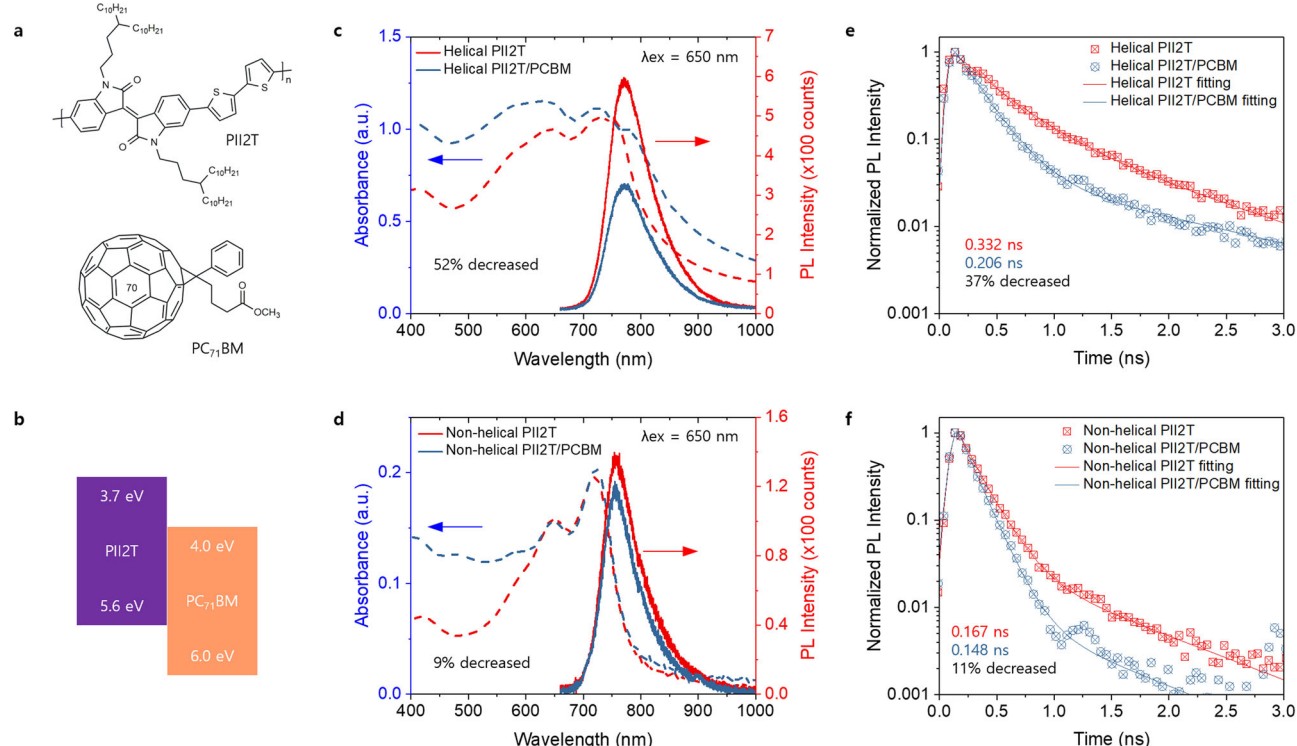

**Fig. 8 Photoluminescence spectroscopy comparing charge generation in helical vs. non-helical bulk heterojunction blends. a** Chemical structures of PII-2T (donor) and PC71BM (acceptor). The helical and non-helical BHJ films were prepared by printing in evaporation and transition regimes, respectively[35]. **b** Energy level alignment of PII-2T and PC71BM. Optical absorbance and photoluminescence spectra of (**c**) the helical pristine and BHJ films and (**d**) the non-helical pristine and BHJ films. Normalized time-resolved photoluminescence spectra of (**e**) the helical pristine and BHJ films and (**f**) the non-helical pristine and BHJ films.

helical structures of semiconducting and conducting polymers will open avenues to exciting optical, electronic, spintronic, mechanical, and biological properties not possible before. Indeed, we demonstrate initial proof-of-concept studies that show helical structures can be beneficial for charge generation in organic solar cells.

## Methods

**Materials**. The isoindigo-based copolymer, PII-2T [number-average MW (Mn) = 30,645 g/mol, weight-average MW (Mw) = 76,809 g/mol, and poly-dispersity index (PDI) = 2.50] was synthesized as previously described[62]. The PII-2T solution was prepared by dissolving the polymer (10–150 mg/ml) in chlorobenzene (CB; anhydrous, 99.8%; Sigma-Aldrich Inc.). The acceptor molecule, [6,6]-Phenyl C71 butyric acid methyl ester (PC71BM) was purchased from Solarmer Materials Inc. A bare Si (0.001–0.005 Ohm.cm, Namkang Hightech) and micro cover glass (VWR Cat No 48366-067) were used as a bottom substrate and top cover, respectively. Corning glass substrates (Fair & Cheer Inc.) were used for spectroscopic studies. The substrates were cleaned with toluene, acetone, and isopropyl alcohol and then blow-dried with a stream of nitrogen to remove contaminants. Poly(sodium 4-styrenesulfonate) (PSS) average Mw ~70,000, powder (Sigma-Aldrich Inc.) was used as a sacrificial layer to transfer the polymer films for TEM characterization.

**Solution-state sample preparation and characterization**. All solution samples except the ones for SAXS were prepared by a drop-and-dry method. The drop-and-dry method is basically concentrating the pristine solution by adding multiple numbers of solution droplets, drying them out and blending with a solution drop. Briefly, a needed concentration was obtained from casting and drying a multiple number of the solution drops on a local spot of the substrate first and blending/shearing with the last drop of the stock solution (2 μl). The sandwiched solution samples were further run through thermal annealing cycles to reach an equilibrium state. We observed the crystalline mesophase beyond a critical concentration both under nonequilibrium droplet-drying conditions and at near-equilibrium conditions after thermal treatments. Selected high concentrated pristine-made solutions (60–150 mg/ml) showed resembling morphologies of crystalline mesophases compared to samples prepared by

the droplet-drying method (Supplementary Fig. 17). This confirms that our droplet-drying method is reliable to study concentration dependent solution phases even without demanding a large amount of materials. For instance, a 50 mg/ml solution was made with dried four drops of 2 μL 10 mg/ml solution on a bare Si substrate and subsequently blending by additional one drop of 2 μL 10 mg/ml solution with a glass coverslip. In order to reach an equilibrium state, the sandwiched sample was run through a moderate heating and cooling process (25 °C → 100 °C → 25 °C) on a Linkam thermal stage (LTS420). The rate of heating and cooling was 5 °C/min. The sandwiched solution was annealed over multiple thermal cycles (2-4 cycles) and equilibrated at room temperature to ensure reaching equilibrium state. The phase after cooling did not evolve any more over time (up to a few days until the solvent was completely evaporated). Additionally, we performed experiments to see whether same sequence of mesophases are formed when the concentrated solution is diluted by adding pure solvent. 0.5 μL pure chlorobenzene was carefully added by capillary force between the sandwiched concentrated solution at desired high concentration. Real-time in-situ microscopy experiments were carried out to preciously examine the phase transition by this dilution method. The birefringence of mesophases was observed using CPOM (Eclipse Ci-POL, Nikon). The samples for spectroscopy were prepared with the same method using corning glass substrates and glass coverslips. UV-vis (Cary 60 UV-Vis, Agilent) spectroscopy was used to calculate the absorption coefficient and investigate polymer conformation in the solution phase. CD spectra were recorded using a JASCO J-810 spectrophotometer. To eliminate contributions from linear dichroism and birefringence, all samples were investigated at numerous sample rotation angles. Identical spectra were recorded at different sample batches demonstrating the chiral nature of the phase.

**Solid-state sample preparation and characterization**. A freeze-drying method was performed to characterize the solution state through electron microscopy tools. The structure of aggregates in solution was imaged using scanning electron microscopy (SEM) and transmission electron microscopy (TEM). The samples sandwiched between a bare Si substrate and a glass coverslip was first submerged in liquid nitrogen. The top coverslip was then removed inside the liquid nitrogen bath. The sample was immediately transferred to a sealed Linkam thermal stage chamber (LTS420) which is held at −100 °C in a nitrogen atmosphere. The temperature was slowly (0.5 °C/min) raised to −80 °C for chlorobenzene sublimation under the vacuum. It took ~6 h to fully sublimate the solvent. Finally,

the temperature was raised to 25 °C. The entire procedure was able to be monitored under the microscope, which ensured us to carefully preserve the solution states. The prepared samples were imaged using SEM (JEOL JSM-7000F at 25 kV accelerating voltage) and TEM (JEOL 2100 Cryo TEM with a LaB$_6$ emitter at 200 kV). For TEM imaging, low electron dose rates ($4–12\,e^-\,Å^{-2}\,s^{-1}$) were applied using spot size 3 to minimize beam-induced alteration. Each image was collected with an exposure time of 1 s, resulting in a dose per image of $4$-$12\,e^-\,Å^{-2}$. A defocus of $-10{,}240$ nm was used throughout all image acquisition to improve contrast. For TEM characterization, all procedure was performed same using PSS layer deposited on the Si substrate. 10 wt% PSS in water solution was spin coated on the Si substrate at 5000 rpm for 1 min. The freeze-drying polymer films on the PSS was transferred on copper grids (Ted pella, 01840-F) in a water bath. Printed PII-2T films were prepared onto PSS-coated substrates by a blade coating method. Briefly, an OTS-treated Si substrate was used as a blade set at an angle of 7°, with a gap of 100 μm between the substrate and the blade. The blade was linearly translated over the stationary substrate while retaining the ink solution within the gap. The PII-2T films were printed on PSS-coated Si substrates at printing speeds at 0.2 mm/s with a substrate temperature of 65 °C. The polymer solutions was 5 mg/ml dissolved in chlorobenzene. The printed films on the PSS was transferred on copper grids in a water bath. The freeze-drying samples prepared on SiO$_2$ were also measured using GIWAXS. GIWAXS measurements were performed at beamline 8-ID-E at the Argonne National Laboratory, with an incident beam energy of 7.35 keV on a 2D detector (PILATUS 1 M) at a 208-mm sample-to-detector distance. Samples were scanned for 10 s in a helium chamber. The x-ray incident angle was set to be above (0.14°) the critical angle (≈0.1°) of the polymer layer (penetration depth, ~5 nm). We note that the GIWAXS analysis of the solution around 200 mg/ml is lacking because of the small sample area (a few hundred micrometers) when compared to the beam irradiated area (~5 mm).

**SAXS experiments and analysis**. SAXS experiments were carried out at the 12-ID-B beamline of the Advanced Photon Source at Argonne National Laboratory using an X-ray beam energy of 13.3 keV. A Pilatus 2 M detector was used primarily at a sample-to-detector distance of 3.6 m. The polymer solution SAXS experiments were performed using a flow cell to prevent beam damage and enable longer exposure times. The flow cell was constructed using a 1 mm diameter quartz capillary connected to PTFE tubing using PTFE heat shrink tubing. The tubing was connected to a syringe pump which cycled the polymer solution at a linear velocity of about 1 mm/s within the capillary while a series of 0.1 s exposures with 3 s delays were accumulated. The isotropic 2D scattering patterns were averaged, reduced, and then background subtracted using the beamline's MATLAB package. The 1D scattering profiles were then analyzed and fit using custom models in SasView.

**PL experiments**. A mixture of 1–1.5 weight ratio for the 10 mg/ml PII-2T (donor) and 15 mg/ml PC71BM (acceptor) was used for helical and non-helical films printed at 0.005 and 0.05 mm/s, respectively. 10 mg/ml and 5 mg/ml of PII-2T solutions were used for neat helical and non-helical films printed at 0.005 and 0.05 mm/s, respectively. The solution concentration was slightly adjusted to present the comparable optical absorption between the neat and blended films, particularly around the maximum optical absorption. The steady-state photoluminescence (PL) and time-resolved photoluminescence (TRPL) spectroscopy measurements were performed in a custom-built experimental setup at the Materials Research Laboratory, University of Illinois.

**MD Simulation**
*System setup*. Topology files of polymers were constructed using Antechamber[63]. The initial coordinate files of the systems were generated using Packmol[64]. The general AMBER force field (GAFF)[65] was used for parameterizing the polymer. The coordinate files of polymers were obtained by drawing and saving as.pdb using PubChem Sketcher[66]. The polymer molecules were solvated in boxes of TIP3P[67] chloroform molecules.

*Simulation and data analysis*. All simulations were run in Amber18[68] with a time step of 2 fs and with hydrogen-containing bonds constrained using the SHAKE algorithm[69]. All production runs were maintained at a constant temperature of 300 K using a Langevin thermostat with a coupling time constant of 2 ps and at a constant pressure of 1 bat using a Berendsen barostat with a τ of 2 ps. A cutoff of 10 Å was used for nonbonded interactions. Frames were saved to trajectory files every 1 ns.

All umbrella sampling simulations were run in Amber18[68]. Amber's harmonic restraints were used for restricting the distance or dihedral angle. In the umbrella sampling using COM distance as a collective variable, we used umbrella windows spaced 0.5 Å apart, and a force constant of 20 kcal/mol Å. The umbrella windows were covering the distance from 1 to 13 Å, and the simulation was run on each window for 10 ns. During the following data analysis, since the distance could not go below 3 Å, that part of the data was cut. In umbrella sampling using dihedral angle as a collective variable, we used umbrella windows spaced 3° apart, and a force constant of 200 kcal/mol rad$^2$. The umbrella windows were covering dihedral angles from $-180°$ to $180°$, and the simulation was run on each window for 2 ns.

Weighted Histogram Analysis Method (WHAM) was used to analyze the data from umbrella sampling simulation and generate the PMF[70,71]. The PMF was generated with an assumed temperature at 300 K and no padding. And the statistical error was estimated using Monte Carlo bootstrap error analysis, with ten fake data sets generated using a random seed of 5. For more details, please see Supplementary Methods.

## Data availability
The authors declare that all data supporting the findings of this study are available within the Article and Supplementary Information files.

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

## Acknowledgements

K.S.P. and Y.D. acknowledge ONR support under Grant No. N00014-19-1-2146, and K.S.P. acknowledges partial support from the Shen Postdoctoral Fellowship. B.B.P. and Y.D. acknowledge NSF DMREF for funding, under Grant Number 17-27605. J.J.K. and Y.D. acknowledge support by the NSF CAREER award under Grant No. 18-47828. J.J.K. also acknowledges partial support from the U.S. Department of Energy, Office of Science, Office of Workforce Development for Teachers and Scientists, Office of Science Graduate Student Research (SCGSR) program. The SCGSR program is administered by the Oak Ridge Institute for Science and Education for the DOE under Contract Number DE-SC0014664. P.K. acknowledges partial support by the NSF MRSEC: Illinois Materials Research Center under Grant Number DMR 17-20633, the 3 M Corporate Fellowship, and the Harry G. Drickamer Graduate Research Fellowship. H.A. and Q.C. acknowledge support from Air Force Office of Scientific Research grant AFOSR FA9550-20-1-0257.

## Author contributions

K.S.P. and Y.D. designed the research project, and Y.D. supervised the project. K.S.P. carried out the experiments and analyzed the data. Z.X. performed the MD simulations and drafted the MD writing under the supervision of D.S. B.B.P developed the freeze-drying set-up with K.S.P. and performed the SEM measurement. H.A performed the TEM measurements and analyzed the data under the supervision of Q.C. J.J.K. performed the SAXS measurements and data fitting and drafted the SAXS writing. P.K. performed the GIXD measurements. K.S.P. and Y.D. wrote the manuscript. All authors discussed, revised, and approved the manuscript.

## Competing interests

The authors declare no competing interests.
