## [Peer Review File · Nature Communications]

Chiral emergence in multistep hierarchical assembly of achiral conjugated polymersEditorial Note: Parts of this Peer Review File have been redacted as indicated to maintain the confidentiality of unpublished data.

REVIEWER COMMENTS

Reviewer #1 (Remarks to the Author):

This paper reports an experimental/simulation-based study of the aggregation and mesophase behavior of a conjugated polymer as a function of concentration. The central claim of the paper is the observation of chiral mesophases using an achiral polymer. It is reminiscent of a string of observations over the past decade of symmetry breaking in molecular liquid crystalline systems, particularly in bent-core mesogen systems and lyotropic chromonic systems. The paper is detailed in its implementation of well-established but start of art characterization techniques that probe the structure of the system at a variety of length scales. This detail is a strength, but the paper is an application of these techniques rather than an advance in them.

There is one key aspect of the paper that I do not understand. In previously reported studies on bent core mesogen systems, the symmetry breaking is local, and chiral domains of opposite handedness are observed with equal probability. If I have understood the authors' claims correctly, they are asserting that they form phases with a unique chirality. I do not understand how this can occur if there is no chirality in their polymer. I wonder if this conclusion made by the authors can be correct. It needs to be resolved before publication. In Figure 5, the author's report molecular-scale helicity in the polymer, but they don't characterize the handedness of the helicity – does it vary from simulation to simulation? If so, then chiral phases may be observed, but they should arise with opposite handedness with equal probability.

Overall, my feeling is that the impact of the paper lies in the significance for the field of organic electronics. In this respect, the potential impact of the paper would be increased by showing how the mesophases encountered lead to functional properties of relevance to organic electronics. The authors allude to applications, but don't really address it head on. Despite the exhaustive characterization, the mesophases have been observed before in systems that are not so far removed from the conjugated oligomers used in this system (e.g., bent-core mesogens), thus I don't view the study as being a broad conceptual advance for the field of organic/lyotropic systems.

In addition to the points above, I made the following observations:

1. The authors interpret the electron micrographs as twisted bundles of fibers in figure 2a and b (e.g., 100 mg/ml). That may be correct, but it is difficult to reach that conclusion based on inspection of Figure 2.
2. Other aspects of the interpretation of the scattering results are plausible, but not proven. See text on page 11 regarding the origin of the increase in the lamellar spacing. The explanation is speculative.
3. Are the phases reported in this paper equilibrium phases? I did not notice the authors address the topic – for example, is the same sequence of mesophases observed on dilution? The arguments presented regarding the formation of the phases are thermodynamic arguments.
4. The text on page 20 is confusing. At line 429, the authors refer to chiral symmetry breaking, but they don't discuss the handedness of the chirality. This is a key issue.

In the end, if this is a paper about local symmetry breaking, and formation of chiral phases of opposite handedness with equal probability, then it seems somewhat routine for soft matter science.

Reviewer #2 (Remarks to the Author):

This manuscript shows how chiral structures emerge from conjugated polymers even though the polymers lack a chiral center. The work provides an incisive description of how self-assembly proceeds from solution towards the solid state, and therefore adds much knowledge to the field. The report of chiral structures also opens the door for new applications of these materials. There are just a few points the authors should address prior to publication.

1. Multiple times the authors claim that this is the first report of assembly into a chiral structure from an achiral conjugated polymer. But, in Ref 33 (Sci Adv 2019) the authors show the same polymer undergoing chiral to achiral transitions under flow. The authors need to clearly delineate how this work differs from their previous study, and likely remove their claims about the first report of chirality.

2. In the GIWAXS work, the loss of texturing at higher concentrations is certainly consistent with helicity arising, but the authors should also mention the possibility of crystals forming in solution and being deposited in random orientations.

3. The simulations provide some computational support for the report of chirality, but fall somewhat short of describing the physics of how chirality arises. In Figure 5c, the asymmetry of dihedral distributions of isolated chains could simply be a result of sampling problems in the simulations. The authors should show in the SI how this dihedral distribution evolves over simulation time.

4. In addition, the authors should analyze the correlations between adjacent dihedral angles, or even next-nearest neighbor dihedral angles. If helicity is indeed present, the authors should see a correlation in adjacent dihedral angles. This should be done both for isolated chain and dimer (two chain) simulations.

5. There is a large discrepancy between the observed pi-stacking distance of about 3.6 angstroms to the free energy minimum of 4.6 angstroms. This is problematic, because the larger spacing is more likely to accommodate twisting of the chains (and therefore allow for helicity). The tighter packing of chains seen in experiment from GIWAXS should essentially allow very little if any twisting of the chains. Most likely this is indicative that the force-fields used are not appropriate for this polymer.

6. There are various symbols that do not show properly in the pdf, such that there are squares in SI as well as question marks in the main text on pg 12, and squares in the main text on pgs 17-18.

Reviewer #3 (Remarks to the Author):

The manuscript by the Ying Diao group describes the measurement of chiral assemblies in solids formed from achiral conjugated polymers. The work is of strong general interest, well-matched to the scope of this fine journal, and the evidence provided is robust and comprehensive. Scholarly presentation is exemplary, with some highlights being the GIXD analysis and discussion, and some beautiful microscopy work. There are some important lingering questions, but this is the nature of science, and the work as submitted is certainly a complete narrative. I recommend acceptance with some minor issues the authors might tackle before publication.

- The authors use the term "pre aggregate" frequently, but this is an ill-defined jargon word of the organic semiconductor community and I recommend against its use. It is not generally clear what the affix "pre" refers to. In organic electronics jargon, "pre" in pre aggregate typically refers to "before the casting process," and the term is meant to refer to solids that form but somehow remain dispersed in

organic semiconductor formulations. And, the word “aggregate” is also poorly defined, and I note that this is a manuscript where the authors also discuss the scientifically well-defined H- and J-aggregates in the optical sense. There are more careful or precise words for such solids. It is reasonable for the authors to refer to these as “dispersed fibrils” because that describes only their shape and their state. Later, the fibrils are found to be nanocrystals (possibly defective ones as organic semiconductors often are). This language would be clearer to a general audience than using “pre aggregate.”

- A central concern of the manuscript is that the authors apply many techniques that can only be used in the absence of liquids, such as electron microscopies which are a centerpiece of the characterization. But most of the evidence for chirality is in solutions, most notably the CD measurements. To ensure the relevance of other measurements of the solids, could the authors provide CD measurements of the solids that they studied, confirming that chirality remains after drying? It is expected that such measurements may not be as low-noise as the solutions, but if they were included in the supplemental information it would alleviate concerns about untracked transitions during drying.

- Related to this question, the SAXS appears to quantify a significant amount of free polymer in the solutions. Could the authors quantify the amount of free polymer relative to dispersed fibrils? Where is the free polymer expected to reside in the dried samples?

- The authors assert that “the mesophases are colloidal, not molecular LCs” [p9] because “the LC mesophases are comprised of pre-aggregated polymer fibers, rather than single polymer chains.” But there is no way to prove that the chain arrangement did not precede crystallization, without studying the solidification evolution (as this group has expertly done before), to show that the mesophases emerge only upon crystallization.

- The optical discussion is notably less rigorous than the other characterization, and there are many assertions in that section that would require more evidence or citation, particularly the assignment of the 715 nm and 650 nm peaks, and discussion that they “can be considered as contributions from J- and H-aggregation” [p14]. Are the authors asserting that these peaks are formally J- and H-aggregates? If so, it seems that far more proof would be required. Consider Sarbu et al., *Journal of Materials Chemistry C* 2015, 3 (6), 1235–1242. <https://doi.org/10.1039/C4TC02444C>, showing multiple isosbestic points with clear transitions in aggregate character. There is also a question of whether any of the features has vibronic character.

- The MD work shows that the molecule has a tendency toward a twisted molecular contour. What it doesn't show (or is not obvious to this reviewer) is why there would be a preference for handedness over a racemic mixture. The molecules are C₂V symmetric. To my mind, this means there should be an equal likelihood of right-hand vs. left-hand twist at the molecular or aggregate level. The statements in the manuscript related to the MD results are somewhat misleading in the sense that they seem to depict the results of the simulation as indicating the origin of the handedness, when instead the MD results are just illustrating that a handedness does develop in the simulation. For example, the Figure 5c dihedral potential is described as “unbalanced on each side of the zero degree (Fig. 5c)... this indicates an asymmetry in the dihedral potential...” [p16] which is an incorrect statement. That plot is a histogram of the dihedrals experienced through the time coordinate (coupled to all the other molecular motions, which are tending toward a twist), not a report of the actual dihedral potential surface. An asymmetry is experienced during the sim, but it doesn't mean that the same asymmetry would develop every time.

- This question brings me to my last point, which is that, given the C₂V symmetry of the molecule, is it possible that a random preferential handedness of any of these phases is “chosen” during the phase transition in which it forms? If the simulation in Figure 5 were done 1000 times with subtly different

starting conditions, would the asymmetry always be the same, or would it have a 50/50 chance of preferring -90° instead of $+90^\circ$? Moreover, if the experiment to make the LC phases were done 1000 times, is it possible that the mesophases at 100 and 140 mg/ml might sometimes show right-handed rather than left-handed helical aggregation? Could the mesophase at >200 mg/ml ever show left-handed helical aggregation? I note that the experimental proof is a far more difficult question to probe, because erasing any handedness from the system may be very, very difficult. In principle, a single molecule having helical handedness could have a prion-like effect (like causing mis-folded proteins) of “seeding” or biasing the morphology toward that same handedness. But this is the only explanation that makes sense to this reviewer in consideration of the molecular symmetry. The idea that a single randomly-chosen handedness quickly becomes dominant makes perfect sense from an LC point of view. It is easy to imagine the less dominant chirality being quickly transformed due to the lower free energy of having uniform handedness, probably via interactions such as that depicted in Figure 6. In my view admitting this possibility takes nothing away from the exciting results in the manuscript, and potentially provides a fuller explanation than the current narrative which leaves the origin of the handedness unanswered.

Point-by-Point Response for Reviewers' Comments in "Chiral emergence in multistep hierarchical assembly of achiral conjugated polymers"

Reviewer comments in blue, corresponding response below in black. Changes made to the manuscript and supplementary information (SI) indicated at the end of each comment.

Reviewer 1 Comments

This paper reports an experimental/simulation-based study of the aggregation and mesophase behavior of a conjugated polymer as a function of concentration. The central claim of the paper is the observation of chiral mesophases using an achiral polymer. It is reminiscent of a string of observations over the past decade of symmetry breaking in molecular liquid crystalline systems, particularly in bent-core mesogen systems and lyotropic chromonic systems. The paper is detailed in its implementation of well-established but start of art characterization techniques that probe the structure of the system at a variety of length scales. This detail is a strength, but the paper is an application of these techniques rather than an advance in them.

There is one key aspect of the paper that I do not understand. In previously reported studies on bent core mesogen systems, the symmetry breaking is local, and chiral domains of opposite handedness are observed with equal probability. If I have understood the authors' claims correctly, they are asserting that they form phases with a unique chirality. I do not understand how this can occur if there is no chirality in their polymer. I wonder if this conclusion made by the authors can be correct. It needs to be resolved before publication. In Figure 5, the author's report molecular-scale helicity in the polymer, but they don't characterize the handedness of the helicity – does it vary from simulation to simulation? If so, then chiral phases may be observed, but they should arise with opposite handedness with equal probability. Overall, my feeling is that the impact of the paper lies in the significance for the field of organic electronics. In this respect, the potential impact of the paper would be increased by showing how the mesophases encountered lead to functional properties of relevance to organic electronics. The authors allude to applications, but don't really address it head on. Despite the exhaustive characterization, the mesophases have been observed before in systems that are not so far removed from the conjugated oligomers used in this system (e.g., bent-core mesogens), thus I don't view the study as being a broad conceptual advance for the field of organic/lyotropic systems.

Response: We thank the reviewer for providing a very insightful assessment of our work. We agree with the reviewer that “the impact of the paper lies in the significance for the field of organic electronics.” and that it is important to determine whether there is local or global symmetry breaking. We address these critical comments head-on in the revised manuscript. We performed additional experiments and clearly showed *for the first time* that helical aggregation of conjugated polymers significantly improves charge generation in bulk heterojunction organic solar cells (**Figure R2** below). This is consistent with our hypothesis that helical aggregation facilitates electronic coupling with electron acceptors at the donor-acceptor interface by better exposing pi-electrons compared to planar, non-chiral counterparts. These new results are presented in paragraphs below. In addition, we further analyzed the simulated molecular helices and determined the handedness. The simulation shows the polymer has almost equal probability to form helix in both handedness (**Figure R3** below). That means the global symmetry breaking observed does not originate from helical conformation of individual, solubilized polymers, but *emerged during LC phase transitions* in a stochastic fashion. The detailed symmetry breaking mechanism supported by additional experimental data is summarized below. Lastly, we would like to stress how our work differentiates from recent literature on bent-core mesogen systems and lyotropic chromonic systems. First, no chiral twist-bent liquid crystals of *polymers* have been observed so far. Past experimental literature all focus on small molecules or colloids. Second, the symmetry breaking mechanism in our work is different from past reports, where flexible, correlated dihedrals and staggered intermolecular stacking give rise to chiral helical polymer fibrils that form the basis for global symmetry breaking. In contrast, previously reported bent-core mesogen requires an overall bent shape to induce local symmetry breaking. Third, the hierarchically assembled structure reported in our work exhibits intriguing complexity, featuring multiple levels of helicity from ~10 nm, ~100 nm, 1 μm to 10 μm scales which is rarely observed before. The complex assembly pathway leads us to discover several new liquid crystal phases not previously known to conjugated polymers. We now present the aforementioned new experiments and simulation studies below.

Charge generation properties. Evaluating the functional property of the helical structure developed through the multistep assembly shows that it is beneficial for charge generation in bulk heterojunction (BHJ) organic solar cells (donor and acceptor structures and energy level alignment shown in Figure R2 a-b). Firstly, the helical film and its counterpart,

non-helical film were prepared by controlling printing regime as reported in our recent work¹. The film printed at the evaporation regime (0.005 mm/s) sustains the originally twisted polymer backbone, leading to helical twisted morphology. In the transition regime (0.05 mm/s), the printing flow induces the planarization of the polymer backbone and thus non-helical aligned morphology. The circular dichroism (CD) spectroscopy confirmed the helical and non-helical characteristics of the printed BHJ films (Figure R1). The steady-state photoluminescence (PL) (Fig. R2, c-d with the axis on the right) was about 52% decreased for the helical BHJ system whereas about 9% decreased for the non-helical BHJ system when compared to each neat polymer film. In the time-resolved PL (Fig. R2, e-f), the decay time of radiative process (fluorescence) for the helical BHJ film is about 37% decreased whereas it is about 11% decreased for the non-helical BHJ system. A clear decrease in PL decay time of the helical BHJ system indicates an increase in energy transfer efficiency between the PII-2T donor polymer and the fullerene acceptor, in line with the steady-state spectral observation. We further observed a slower PL decay in the helical neat polymer film (0.332 ns) than the non-helical neat films (0.167 ns), suggesting a longer singlet exciton lifetime² in the helical film. This result shows the helical system can provide a higher probability for excitons to reach the donor-acceptor interface for electron-hole dissociation. Together, steady state and time-resolved PL results suggest that helical structure of donor polymers leads to a longer exciton lifetime and more efficient charge splitting at the donor-acceptor interface compared to non-helical structures. Both factors combined result in improved charge generation in bulk heterojunction organic solar cells.

Figure R1. CD spectra of the helical (left) and non-helical (right) PII-2T/PC71BM BJJ films, confirmed the chiral and non-chiral characteristics preserved after blending with PC71BM. CD measurements were carried out with the sample rotated in-plane 0° and 90° angles to rule out the linear dichroism and birefringence.

Figure R2. Photoluminescence spectroscopy comparing charge generation in helical vs. non-helical bulk heterojunction blends of PII-2T polymer (donor) and PC71BM (acceptor). (a) Chemical structures of PII-2T and PC71BM. The helical and non-helical BJJ films were prepared by printing in evaporation and transition regimes, respectively¹. (b) Energy level alignment of PII-2T and PC71BM. Optical absorbance and photoluminescence spectra of (c) the helical pristine and BJJ films and (d) the non-helical pristine and BJJ films. Normalized time-resolved photoluminescence spectra of (e) the helical pristine and BJJ films and (f) the non-helical pristine and BJJ films.

MD simulation for molecular handedness. Next, we have further assessed the handedness of the molecular helicity by MD simulations. Figure R3(a) shows an example of the PII-2T 30mer backbone formed in a right-handed helix as in the relaxed structure at the start of simulation. Figure R3(b, c) show the two observed helices captured during the simulation, which show the right- and left-handed helix, respectively. Figure R3(d) shows the histogram of helix handedness counted from about 3000 frames, showing the polymer has almost equal probability to form helix in both handedness. Therefore, we conclude that there is no intrinsic tendency for the isolated polymer to form helices with one dominant handedness. The simulations here provide evidence that conformations with both handedness exist even at a single polymer level. So what explains global symmetry breaking when the chiral mesophases formed? We propose that handedness of the mesophase is stochastically “chosen” during the phase transition. In a racemic solution of chiral helical polymer fibrils, a population bias towards a certain handedness can transiently exist due to stochastic fluctuations. At the time when the polymer fibrils coalesce / nucleate into a chiral mesophase, such population bias can be amplified through conversion of the minority into the majority handedness, driven by intermolecular interactions depicted in Figure 6 and free energy minimization when forming a coherent mesophase with uniform handedness. This view is supported by the observed stochastic nature of chiral symmetry breaking – the twist-bent mesophases I, II and striped twist-bent mesophase can all adopt both handedness with certain probabilities. According to our newly performed CD experiments examining 50 samples for each mesophase, we observed left-handed twist-bent mesophase I and II form at 56% and 62% probability respectively, and right-handed striped twist-bent mesophase form at 78% probability. This new dataset reveals that bias towards certain handedness increases with increasing volume fraction/concentration of the polymer in solution. This suggests that asymmetric intermolecular interactions may play an important role in chiral symmetry breaking, such as asymmetric staggered stacking shown in Figure 6. Such asymmetry in intermolecular interactions may be amplified as the polymers pack closer in a mesophase.

Figure R3. (a) Backbone of PII-2T 30mer showing a right-handed helix in the relaxed structure at the start of simulation. Captured examples of the right- (b) and left-handed (c) helix formed during simulation. (d) Fraction of frames that show either left- or right-handedness in total counts of frames showing helicity.

Changes made to the manuscript:

- 1) We included the Figure R1 and R2 in Supplementary Information (Fig. S16) and the main manuscript (Fig. 8), respectively. Also the related description was added in the main manuscript (page 24-25).
- 2) We included the Figure R3 in Supplementary Information (Fig. S13) and changed the related description in the main manuscript (page 16-17).

In addition to the points above, I made the following observations:

1. The authors interpret the electron micrographs as twisted bundles of fibers in figure 2a and b (e.g., 100 mg/ml). That may be correct, but it is difficult to reach that conclusion based on inspection of Figure 2.

Response: Due to a high density of fibers in the samples, it might be difficult to distinguish the twistiness. For better visualization we provide additional images taken from the freeze-dried PII-2T mesophase at 100 mg/ml. The raw TEM images and the corresponding processed images using ImageJ show the twisted fibers traced with the yellow dotted lines.

Figure R4. TEM images of freeze-dried PII-2T mesophase at 100 mg/ml. The yellow dotted lines in images exhibits the twisted structures. Further imaging analysis was performed using ImageJ.

Changes made to the manuscript:

We included the Figure R4 in Supplementary Information (Fig. S5) and mentioned it in the main manuscript (page 8).

2. Other aspects of the interpretation of the scattering results are plausible, but not proven. See text on page 11 regarding the origin of the increase in the lamellar spacing. The explanation is speculative.

Response: We thank the reviewer for raising this question. We are very confident with the conclusion from SAXS as we have definitively pinpointed that the structure factor at high q comes from lamella stacking, using two different modeling approaches we developed. Please refer to pre-print article for more details about the SAXS modeling approaches, <https://doi.org/10.26434/chemrxiv-2021-knxg8> (we received very positive referee reports and have been invited to submit a revision). With regard to the origin of lamella spacing increase, linking increase of lamella spacing with increased backbone torsion is consistent with UV-Vis absorption spectra of PII-2T mesophases (Figure 4 in the main manuscript). The decrease in absorbance ratio (0-0)/(0-1) and absorption coefficient as well as the blue-shifted (0-0) and (0-1) peaks with increasing solution concentration suggest a decrease in π -conjugation and increase in backbone torsion from isotropic to twisted mesophase. A schematic drawing shown in Figure R5 illustrates the possible structural transition. With the compression along the molecular (fiber) long axis due to the increased twistiness, the effective volume of polymer chains may be increased, leading to increased lamellar stacking distance.

Figure R5. Schematic illustration of lamellar stacking distance increase as the polymers are more twisted with change increasing solution concentration.

Our explanation is further supported by previous observations that helical structures lead to increased interchain spacing. For instance, Wang and Swager et al.³ reported the high magneto-optic activity observed in a chiral helical poly-3-(alkylsulfone)thiophene created by the addition of chiral center on the side chain. Grazing-incidence wide-angle X-ray scattering patterns revealed the (100) interchain spacing (lamella spacing) of 23.5-25.6 Å for helical structures, which is considerably larger than the lamella spacing (16.4-21.5 Å) of simple non-chiral poly-3-(alkyl)thiophenes with C4 to C10 carbon chains that organize in lamellar

structures in solution cast films. Another work reported by Davidson and Segalman et al.⁴ is controlling the self-assembly of block copolymers with variable chain shape and stiffness. The small-angle X-ray scattering showed that the block copolymer with the helical block displays a significantly larger domain (approximately a 20% increase in domain size) than the block copolymer with the non-helical chain. This observation was attributed to the unfavorable packing interactions from the helical block in the confined cylindrical core.

Changes made to the manuscript:

We added the aforementioned references [3-4] to support our explanation in the main manuscript (page 11).

3. Are the phases reported in this paper equilibrium phases? I did not notice the authors address the topic – for example, is the same sequence of mesophases observed on dilution? The arguments presented regarding the formation of the phases are thermodynamic arguments.

Response: We thank the reviewer for raising this insightful question. We confirm that the phases reported in this study are equilibrium phases by both thermal annealing experiments and dilution experiments suggested by the reviewer. All the sandwiched solution samples were run through thermal annealing cycles (25 °C→100 °C→25 °C) to reach an equilibrium state. The rate of heating and cooling was 5 °C/min. The sandwiched solution was annealed over multiple thermal cycles and equilibrated at room temperature to ensure the state of the solution no longer changes. Figure R6 shows an example showing the equilibrium phase with nematic tactoids consistently observed in 60 mg/ml PII-2T solutions over the multiple thermal annealing cycles. The phase after cooling did not evolve any more over time (up to a few days). Additionally, we observed the same sequence of mesophases when the concentrated solution was diluted by adding pure solvent. The striped twist-bent mesophase turns to twist-bent II and I phases with decreasing the solution concentration upon dilution. Starting from twist-bent mesophase II, we observed twist-bent mesophase I by dilution. Nematic tactoids were observed at the border of mesophases when diluting twist-bent mesophase I and II, possibly due to the fact that it is easier for the droplets to break off from a continuous mesophase at the phase boundaries. The isotropic phase (dark green) is also seen at the surrounding area of the crystalline mesophases.

Figure R6. Cross-polarized optical microscope images of 60 mg/ml PII-2T solution as-prepared and after subsequent thermal cycles. Initially, non-equilibrium liquid crystalline phase is shown in the glass-sandwiched solution sample. After thermal cycles, equilibrium phase with nematic tactoids is observed.

Changes made to the manuscript:

We added the description that the phase is equilibrium state in page 5 and also described in detail how to obtain the stable phase in the section of Materials and Methods.

4. The text on page 20 is confusing. At line 429, the authors refer to chiral symmetry breaking, but they don't discuss the handedness of the chirality. This is a key issue.

Response: We thank the reviewer for raising this important point. In isotropic solutions, single polymer chain has equal probability to form helix in both handedness according to newly added MD simulation results. At higher concentrations when mesophases form, CD measurements for the twisted mesophases show that both left- and right-handedness can occur but not with an equal probability. Examining 50 samples for each mesophase, we observed left-handed twist-bent mesophase I and II form at 56% and 62% probability respectively, and right-handed striped twist-bent mesophase form at 78% probability. This new dataset reveals that bias

towards certain handedness increases with increasing volume fraction/concentration of the polymer in solution. This suggests that asymmetric intermolecular interactions may play an important role in chiral symmetry breaking, such as asymmetric staggered stacking shown in Figure 6. Such asymmetry in intermolecular interactions may be amplified as the polymers pack closer in a mesophase.

Changes made to the manuscript:

We have revised pages 16-20 and to now include description of handedness throughout the paragraph and further added a paragraph discussing the symmetry breaking mechanism.

In the end, if this is a paper about local symmetry breaking, and formation of chiral phases of opposite handedness with equal probability, then it seems somewhat routine for soft matter science.

Response: We would like to emphasize that the polymer chiral mesophase formation in this study is caused by the global symmetry breaking rather than local symmetry breaking, owing to long-range coherent inter-fiber interactions that leads to a single handedness in twist-bent mesophases I, II and striped twist-bent mesophase. Our findings are unique and have not been reported before, as twist-bent mesophases have not been experimentally observed in polymer systems. Here, we not only discover several previously unknown chiral mesophases but further report the origin of chiral emergence in conjugated polymers and uncover their complex hierarchical morphology. Given that conjugated polymers is one of the most important classes of electronic materials, this finding can lead to new electronic materials leveraging their unique optical, optoelectronic, spintronic, mechanical and biological properties not possible before in absence of chirality. Indeed, we added new results that these unique structures are beneficial for charge generation in organic solar cells.

Reviewer 2 Comments

This manuscript shows how chiral structures emerge from conjugated polymers even though the polymers lack a chiral center. The work provides an incisive description of how self-assembly proceeds from solution towards the solid state, and therefore adds much knowledge to the field. The report of chiral structures also opens to the door for new applications of these materials. There are just a few points the authors should address prior to publication.

Response: We greatly appreciate the positive comments on our work. We have fully addressed the comments detailed below.

1. Multiple times the authors claim that this is the first report of assembly into a chiral structure from an achiral conjugated polymer. But, in Ref 33 (Sci Adv 2019) the authors show the same polymer undergoing chiral to achiral transitions under flow. The authors need to clearly delineate how this work differs from their previous study, and likely remove their claims about the first report of chirality.

Response: We agree with the reviewer on removing this claim. In our previous work (Sci Adv 2019), we did briefly bring up a twist-bent chiral liquid crystal of conjugated polymers to explain the impact of printing flow on polymer conformation and assembly. However, in this new work we not only discover several previously unknown chiral mesophases, we further report the origin of chiral emergence in conjugated polymers and uncover their complex hierarchical morphology. All these new findings represent a major leap from previous literature reports.

Changes made to the manuscript:

We removed the term, “for the first time” from the abstract, page 3 and 26.

2. In the GIWAXS work, the loss of texturing at higher concentrations is certainly consistent with helicity arising, but the authors should also mention the possibility of crystals forming in solution and being deposited in random orientations.

Response: We thank the reviewer for raising this question. We anticipate loss of texturing in both in-plane and out-of-plane directions if the crystals formed in solution and being deposited in random orientations during the process. However, according to the SEM and TEM images

(Figure 2a, 2b) of the concentration series samples that were also used to perform in GIWAXS, we observed strong and uniform in-plane texturing and alignment, while loss of out-of-plane texturing. This suggests that loss of out-of-plane texturing is due to internal structure of the aggregates that constitute the mesophase. Furthermore, in TEM and SEM we did not observe two different types of crystal aggregates, aside from the one already aggregated and constitutes the mesophase.

Changes made to the manuscript:

We shortly discussed about the loss of out-of-plane texturing in GIWAXS in page 12.

3. The simulations provide some computational support for the report of chirality, but fall somewhat short of describing the physics of how chirality arises. In Figure 5c, the asymmetry of dihedral distributions of isolated chains could simply be a result of sampling problems in the simulations. The authors should show in the SI how this dihedral distribution evolves over simulation time.

Response: We agree with the reviewer's contention that the simulations have helped us in generating potential hypotheses for emergence of chirality but fall short of describing the complete physics of the process. Long timescale simulations of multimers could have provided the much-needed mechanistic explanations but these simulations remain computationally intractable with molecular details. To address the issue of limited sampling, we have now obtained the dihedral frequency distribution for the simulations for a single monomer performed over long timescale. Figure R7 shows representative frequency distribution plots of six thiophene-thiophene dihedral angles along various point of the multimer at five different time points in simulation (52.4, 104.8, 157.2, 209.6 and 262 ns). From this figure, we conclude that the imbalanced distribution is intrinsic to the polymer and not caused by the limited sampling. We propose that the imbalanced distribution could lead to chirality. Due to the rapid fluctuations of the single polymer chain in solution and absence of any stabilizing interactions from other polymeric chains, the single chain does not form stable chiral structures but in assembled multi-chain solution the existence of other chains could stabilize a chiral conformation as seen in the experimental results.

Figure R7. Frequency distribution plots showing growth of thiophene-thiophene dihedral angles over time and the intrinsic unbalanced tendency of dihedrals. (a) T-T dihedral between the 3rd and 4th monomer; (b) T-T dihedral between the 9th and 10th monomer; (c) T-T dihedral between the 14th and 15th monomer; (d) T-T dihedral between the 19th and 20th monomer; (e) T-T dihedral between the 24th and 25th monomer; (f) T-T dihedral between the 29th and 30th monomer.

Changes made to the manuscript:

We added the Figure R7 in Supporting Information, Figure S14 to demonstrate that the asymmetry of dihedral distributions of isolated chains is not due to the sampling problems in the simulations.

4. In addition, the authors should analyze the correlations between adjacent dihedral angles, or even next-nearest neighbor dihedral angles. If helicity is indeed present, the authors should see a correlation in adjacent dihedral angles. This should be done both for isolated chain and dimer (two chain) simulations.

Response: Thanks to the referee for this excellent point. We have analyzed the correlations between adjacent dihedral angles in helical regions. In the following, the same abbreviations for the polymer repeating units as in the manuscript are used here: I for isoindigo and T for thiophene. We found that the I-I dihedral angle is the most rigid, with the narrowest distribution centered around 25° (Supplementary Information, Fig. S9). The I-T and T-I dihedral angles

fluctuate between four well-defined positions (two cis and two trans conformations), all deviating from co-planarity by $\sim 30^\circ$. Here we have further investigated the adjacent T-T dihedral pairs because (1) it is the most fluctuated among the backbone dihedrals and (2) its cis-conformation is a characteristic of the curved region shown in Fig. 5(d) of the main manuscript. Through analysis on adjacent T-T dihedrals from simulation, we have found the time window in which two adjacent T-T dihedrals are highly correlated (Pearson's $r > 0.7$). Furthermore, the formation of helix requires correlated motion of multiple adjacent dihedrals. However, the helical structures in the simulations fluctuate significantly and their pitch length also varies which makes it harder to find strong correlation between multiple dihedrals. We believe that the correlation would be stabilized by surrounding multiple chains in solution during experiment instead of a single chain in simulation.

Figure R8. (Top) Selected time-dependent dihedral angle plots of the adjacent T-T pair (yellow and blue); (left) a pair of 3rd-4th and 4th-5th position, (middle) a pair of 8th-9th and 9th-10th position and (right) a pair of 23th-24th and 24th-25th position. (Bottom) Dihedral angle distribution of each adjacent T-T pair mentioned above.

Changes made to the manuscript:

We added the Figure R8 in Supporting Information, Figure S12 to demonstrate that adjacent T-T dihedrals in helical regions are highly correlated.

5. There is a large discrepancy between the observed pi-stacking distance of about 3.6 angstroms to the free energy minimum of 4.6 angstroms. This is problematic, because the larger spacing is more likely to accommodate twisting of the chains (and therefore allow for helicity). The tighter packing of chains seen in experiment from GIWAXS should essentially allow very little if any twisting of the chains. Most likely this is indicative that the force-fields used are not appropriate for this polymer.

Response: We thank the reviewer for raising this question. The generalized AMBER force field (GAFF) we used for our polymer is valid as the free energy minimum of 4.6 Å closely matches with the average π -stacking distance (~ 4.4 Å) in amorphous regions of the films measured by GIWAXS. Also we would like note that the GAFF is the most commonly used for many conjugated polymer studies⁵⁻⁷. The π -stacking distance of about 3.6 Å obtained from GIWAXS may correspond to domains from crystallization of dispersed polymer chains during the freeze-drying process. We infer that the π -stacking distance internal to the polymer fibrils is close to ~ 4.4 Å; this diffraction signature is overlapped with the amorphous ring in GIWAXS so it is very challenging to differentiate from one another. Therefore, we resorted to TEM electron diffraction on single polymer fibers to confirm our inference. Figure R9 shows the TEM image and corresponding diffraction of PII-2T solution dispersed fibrils, indicating that the π -stacking distance is ~ 4.37 Å, rather than 3.6 Å coming from solution dispersed polymer fibrils. Since TEM directly probes stacking associated with the polymer fibers, and exclude regions where crystallization of dispersed polymer chains may occur, this result support our inference from GIWAXS. Further, this result is in line with the MD simulation result and our previous GIWAXS study¹, where the diffraction signature was appeared at $q = \sim 1.42$ Å⁻¹ ($d = 4.42$ Å) and $\chi = \pm 7 - 15^\circ$ for helical aggregates in printed films. Accordingly, we have significantly revised the GIWAXS analysis and discussion based on your suggestions and additional experimental observation.

[FIGURE REDACTED]

Changes made to the manuscript:

We added the discussion about GIWAXS π -stacking on page 12-13 and slightly changed the Figure 3 with lamellar stacking orientation.

6. There are various symbols that do not show properly in the pdf, such that there are squares in SI as well as questions marks in the main text on page 12, and squares in the main text on pages 17-18.

Response: We thank the reviewers to point out these errors. We have adequately fixed these errors now.

Reviewer 3 Comments

The manuscript by the Ying Diao group describes the measurement of chiral assemblies in solids formed from achiral conjugated polymers. The work is of strong general interest, well-matched to the scope of this fine journal, and the evidence provided is robust and comprehensive. Scholarly presentation is exemplary, with some highlights being the GIXD analysis and discussion, and some beautiful microscopy work. There are some important lingering questions, but this is the nature of science, and the work as submitted is certainly a complete narrative. I recommend acceptance with some minor issues the authors might tackle before publication.

Response: We greatly appreciate the positive comments on our work. We have fully addressed the comments detailed below.

1. The authors use the term “pre aggregate” frequently, but this is an ill-defined jargon word of the organic semiconductor community and I recommend against its use. It is not generally clear what the affix “pre” refers to. In organic electronics jargon, “pre” in pre aggregate typically refers to “before the casting process,” and the term is meant to refer to solids that form but somehow remain dispersed in organic semiconductor formulations. And, the word “aggregate” is also poorly defined, and I note that this is a manuscript where the authors also discuss the scientifically well-defined H- and J-aggregates in the optical sense. There are more careful or precise words for such solids. It is reasonable for the authors to refer to these as

“dispersed fibrils” because that describes only their shape and their state. Later, the fibrils are found to be nanocrystals (possibly defective ones as organic semiconductors often are). This language would be clearer to a general audience than using “pre aggregate.”

Response: We agree with the reviewer that the term “pre aggregate” may not be suitable to define the polymer fibers in our systems.

Changes made to the manuscript:

We have revised our wording accordingly on page 1, 4, 8, 9, 11, 12, 14 and 26, instead using “dispersed polymer fibrils”.

2. A central concern of the manuscript is that the authors apply many techniques that can only be used in the absence of liquids, such as electron microscopies which are a centerpiece of the characterization. But most of the evidence for chirality is in solutions, most notably the CD measurements. To ensure the relevance of other measurements of the solids, could the authors provide CD measurements of the solids that they studied, confirming that chirality remains after drying? It is expected that such measurements may not be as low-noise as the solutions, but if they were included in the supplemental information it would alleviate concerns about untracked transitions during drying.

Response: We thank the reviewer for this thoughtful suggestion. The CD spectra of the freeze-dried samples can help validate the chirality preservation from the solution to solid state samples that we have explored in this study. Figure R10 shows the examples of the CD spectra and corresponding optical microscopic images of the freeze-dried solid samples prepared at 100 and 200 mg/ml, respectively. The solid state samples still display the CD signals, indicating that the chirality is persistent after removing the solvent.

Figure R10. Cross-polarized optical microscope images and corresponding CD spectra of the freeze-dried PII-2T prepared at 100 and 200 mg/ml, respectively. CD measurements were carried out with the sample rotated multiple in-plane angles to rule out the linear dichroism and birefringence. The CD spectra show the chirality is preserved after removing the solvent during the freeze drying process.

Changes made to the manuscript:

We included the Figure R10 in Supplementary Information, Figure S3 and also added a sentence to mention this figure on page 8 where we firstly introduce freeze-dried samples.

3. Related to this question, the SAXS appears to quantify a significant amount of free polymer in the solutions. Could the authors quantify the amount of free polymer relative to dispersed fibrils? Where is the free polymer expected to reside in the dried samples?

Response: We thank the reviewer for raising this question. Because of the large size of the dispersed fibrils and the limited q-range of SAXS we can only see the trailing Porod scattering of the dispersed fibrils instead of its entire scattering profile. Therefore, it is not possible to quantify its contribution, even in a relative manner to the free polymer. We expect the free polymers to be present in the surrounding region where no fibrils found in the electron microscopy images. In areas where fibers were not observed, the substrate was coated with a thin layer, but electron microscopy was not able to resolve fine textures in these regions.

4. The authors assert that “the mesophases are colloidal, not molecular LCs” [p9] because “the LC mesophases are comprised of pre-aggregated polymer fibers, rather than single polymer chains.” But there is no way to prove that the chain arrangement did not precede crystallization, without studying the solidification evolution (as this group has expertly done before), to show that the mesophases emerge only upon crystallization.

Response: We agree with the reviewer and have revised the sentences according to the suggestion.

Changes made to the manuscript:

We changed the sentence, “In other words, we infer that mesophases are colloidal, not molecular LCs, validation of which requires directly probing the solidification process using in situ characterization techniques.” on page 9. Also a related ref⁸ was added.

5. The optical discussion is notably less rigorous than the other characterization, and there are many assertions in that section that would require more evidence or citation, particularly the assignment of the 715 nm and 650 nm peaks, and discussion that they “can be considered as contributions from J- and H-aggregation” [p14]. Are the authors asserting that these peaks are formally J- and H- aggregates? If so, it seems that far more proof would be required. Consider Sarbu et al., Journal of Materials Chemistry C 2015, 3 (6), 1235–1242. <https://doi.org/10.1039/C4TC02444C>, showing multiple isosbestic points with clear transitions in aggregate character. There is also a question of whether any of the features has vibronic character.

Response: We thank the reviewer for raising this insightful point. We have significantly revised the UV-Vis analysis and discussion based on your suggestions and recent literature. First of all, the two main peaks around 715 nm and 650 nm in our study indicate they are vibronic characters of the same excitonic transition. Because the difference of those two peaks is about 1400 cm^{-1} (0.17 eV), which is corresponding to the vibrational frequency of the aromatic-quinoidal stretching mode for nearly all π -conjugated molecules⁹. Therefore, we now assign peaks at ~715 nm and ~650 nm as electronic transition (0-0) and its higher order vibronic transition (0-1), respectively. Thanks to the great work developed by Spano and colleagues⁹, characterizing the absorption vibronic peak ratio (A_{0-0}/A_{0-1}) can help identify the state of

excitonic coupling (defined as H and J aggregates) in conjugated polymer systems. Briefly, the ratio is proportional to the conjugation length, which is related to exciton delocalization. J-promoting nature with intra-unit coupling have a much larger 0-0 vibronic peak intensity than all other peaks in the absorption. In contrast, inter-unit coupling (or H-aggregation) promotes higher order vibronic transitions than the lower ones. So the relative intensity of the (0-0) vs. (0-1) peak has been used as an indicator for examining the strength of excitonic interaction and conjugation length (therefore polymer conformation)^{10, 11}.

With this knowledge, we have revised our discussion about the optical transition property of PII-2T mesophases. The peaks around 710 and 655 nm are the (0-0) and (0-1) vibronic transition along the polymer chains, respectively, more likely contributed from both aggregated fiber and dispersed single polymer chains. Clearly, the decrease in the ratio of (0-0)/(0-1) and absorption coefficient (Fig. R 11b) as well as the hypsochromic shift of the (0-0) and (0-1) peak position (Fig. R11c) indicate H-like character enhanced with increasing solution concentration. This suggests that the aggregates are more twisted and/or polymer conformation is more torsional as well.

Figure R11. Optical transition property of PII-2T mesophase. (a) UV-Vis absorption spectra of PII-2T solutions. The arrows indicate peak changes with change increasing solution concentration. (b) The peak ratio, (0-0)/(0-1), absorption coefficient (ϵ) and (c) the (0-0) and (0-1) peak position as concentration increases.

Changes made to the manuscript:

We changed the UV-Vis absorption interpretation (page 14-15) with replacing Figure R11 to Figure 4 in the main manuscript.

6. The MD work shows that the molecule has a tendency toward a twisted molecular contour. What it doesn't show (or is not obvious to this reviewer) is why there would be a preference for handedness over a racemic mixture. The molecules are C₂V symmetric. To my mind, this means there should be an equal likelihood of right-hand vs. left-hand twist at the molecular or aggregate level. The statements in the manuscript related to the MD results are somewhat misleading in the sense that they seem to depict the results of the simulation as indicating the origin of the handedness, when instead the MD results are just illustrating that a handedness does develop in the simulation. For example, the Figure 5c dihedral potential is described as “unbalanced on each side of the zero degree (Fig. 5c)... this indicates an asymmetry in the dihedral potential...” [p16] which is an incorrect statement. That plot is a histogram of the dihedrals experienced through the time coordinate (coupled to all the other molecular motions, which are tending toward a twist), not a report of the actual dihedral potential surface. An asymmetry is experienced during the sim, but it doesn't mean that the same asymmetry would develop every time.

Response: This is an excellent point that we are yet to get to the gist of symmetry breaking. Taken together with point 7, we are in full agreement with the referee's terrific (!) suggestion on the symmetry breaking mechanism as being “chosen” during the phase transitions. The stochastic nature of left- vs. right- handedness for chiral mesophases is indeed observed in experiments. Details are discussed in response to point 7. Below we specifically address the referee's point on the validity of asymmetric torsional potential and its relevance to symmetry breaking during phase transitions.

Indeed, the MD results presented in the original manuscript only shows that chiral helical conformation exists transiently in single molecules in solution. To understand whether backbone dihedrals relate to chiral symmetry breaking, we performed three additional simulations and analyses as below.

1) We validated that the dihedral angle asymmetry is persistent over time. And starting from different conformations, the same dihedral angle asymmetry is still preferred. Specifically, we obtained the dihedral frequency distribution for the simulations for a single monomer performed over the long timescale. Figure R12 shows representative frequency distribution plots of six thiophene-thiophene dihedral angles along various point of the multimer at five different time points in simulation (52.4, 104.8, 157.2, 209.6 and 262 ns). From this figure, we

conclude that the imbalanced distribution is intrinsic to the polymer and not caused by the limited sampling. We further performed five new simulations starting from five different conformations randomly picked from the previous trajectory, and plotted the T-T dihedral change (between the 14th and 15th monomer) shown in Figure R13. These results show that the same dihedral angle asymmetry is preferred despite random starting points.

Figure R12. Frequency distribution plots showing growth of thiophene-thiophene dihedral angles over time and the intrinsic unbalanced tendency of dihedrals. (a) T-T dihedral between 3rd and 4th monomer; (b) T-T dihedral between 9th and 10th monomer; (c) T-T dihedral between 14th and 15th monomer; (d) T-T dihedral between 19th and 20th monomer; (e) T-T dihedral between 24th and 25th monomer; (f) T-T dihedral between 29th and 30th monomer.

Figure R13. Plots of T-T dihedral angles between the 14th and 15th monomer from 5 newly-simulated trajectories started from random frames picked from the first trajectory, showing

the same dihedral angle asymmetry is preferred despite random starting points.

2) We validated that adjacent dihedral angles are highly correlated (Pearson's $r > 0.7$) shown in Figure R14, which help explain why chiral helix form in the first place in a single polymer chain.

Figure R14. (Top) Selected time-dependent dihedral angle plots of the adjacent T-T pair (yellow and blue); (left) a pair of 3rd-4th and 4th-5th position, (middle) a pair of 8th-9th and 9th-10th position and (right) a pair of 23th-24th and 24th-25th position. (Bottom) Dihedral angle distribution of each adjacent T-T pair mentioned above.

3) We determined the probability of left- vs. right- handedness appearing in a single polymer chain. Figure R15(a) shows an example of the PII-2T 30mer backbone formed in a right-handed helix as in the relaxed structure at the start of simulation. Figure R15(b, c) show the two observed helices captured during simulation, which show the right- and left-handed helix, respectively. Figure R15(d) shows the histogram of helix handedness counted from about 3000 frames, showing the polymer has almost equal probability to form helix in both handedness. Therefore, we conclude that there is no intrinsic tendency for the isolated polymer to form helices with one dominant handedness.

In summary, the MD simulations provide evidence that helical conformations with both handedness exist in solution at a single polymer level due to correlated dihedrals along the polymer backbone. However, given almost equal probabilities of forming left- and right-

handed conformations, there is absence of local symmetry breaking at the single polymer level. Nonetheless, weak asymmetric dihedral angle distributions do exist and persist over time. While such weak asymmetry does not seem to bias the handedness at the single polymer level, it could be amplified during multimolecular assembly process when adjacent polymers strongly interact. However, whether such dihedral asymmetry play a role in global symmetry breaking during chiral mesophase formation remains unknown.

Figure R15. (a) Backbone of PII-2T 30mer showing a right-handed helix in the relaxed structure at the start of simulation. Captured examples of the right- (b) and left-handed (c) helix formed during simulation. (d) Fraction of frames that show either left- or right-handedness in total counts of frames showing helicity.

Changes made to the manuscript:

We placed the Figure R12-R15 in Supplementary Information, Figure S14, S15, S12 and S13, respectively. Particularly, Figure R14 (a) and Figure R15 (b-c) are included in Figure 5 (e-f) of the main manuscript to demonstrate each the correlation of adjacent T-T dihedrals in helical regions and a right- and left-handed helix formed during simulation.

7. This question brings me to my last point, which is that, given the C₂V symmetry of the molecule, is it possible that a random preferential handedness of any of these phases is “chosen” during the phase transition in which it forms? If the simulation in Figure 5 were done 1000 times with subtly different starting conditions, would the asymmetry always be the same, or would it have a 50/50 chance of preferring -90° instead of +90°? Moreover, if the experiment to make the LC phases were done 1000 times, is it possible that the mesophases at 100 and 140 mg/ml might sometimes show right-handed rather than left-handed helical aggregation? Could the mesophase at >200 mg/ml ever show left-handed helical aggregation? I note that the experimental proof is a far more difficult question to probe, because erasing any handedness from the system may be very, very difficult. In principle, a single molecule having helical handedness could have a prion-like effect (like causing mis-folded proteins) of “seeding” or biasing the morphology toward that same handedness. But this is the only explanation that makes sense to this reviewer in consideration of the molecular symmetry. The idea that a single randomly-chosen handedness quickly becomes dominant makes perfect sense from an LC point of view. It is easy to imagine the less dominant chirality being quickly transformed due to the lower free energy of having uniform handedness, probably via interactions such as that depicted in Figure 6. In my view admitting this possibility takes nothing away from the exciting results in the manuscript, and potentially provides a fuller explanation than the current narrative which leaves the origin of the handedness unanswered.

Response: We are truly grateful for this very valuable suggestion! We fully agree with the proposed picture that the handedness of the mesophase is stochastically chosen during the phase transition to reach a lower free energy state for having uniform handedness in a long-range ordered mesophase. Indeed, we observed this stochastic nature from MD simulations, where helix of a single polymer with left- and right-handedness forms at almost equal probability (see response to point 6). We also observed this stochastic nature from experiments

at the mesophase level. According to our newly performed CD experiments examining 50 samples for each mesophase, we observed left-handed twist-bent mesophase I and II form at 56% and 62% probability respectively, and right-handed striped twist-bent mesophase form at 78% probability. It remains a question whether the experimentally observed biases towards one handedness in a chiral mesophase relates to asymmetries at the molecular scale, namely asymmetric dihedral angle distribution and asymmetric staggered stacking. Indeed, it is very difficult to erase any handedness from the system in experiments as the referee pointed out. Further, we do acknowledge that our sampling size is limited to reveal true statistics. Nonetheless, this new dataset reveals that bias towards certain handedness increases with increasing volume fraction/concentration of the polymer in solution. This suggests that asymmetric intermolecular interactions may play an important role in chiral symmetry breaking, such as asymmetric staggered stacking shown in Figure 6. Such asymmetry in intermolecular interactions may be amplified as the polymers pack closer in a mesophase.

Changes made to the manuscript:

We now add a paragraph to discuss this symmetry breaking mechanism as the following on page 22.

“What is the symmetry breaking mechanism that underpins formation of chiral mesophases? We propose that handedness of the mesophase is stochastically “chosen” during the phase transition. In a racemic solution of chiral helical polymer fibrils, a population bias towards a certain handedness can transiently exist due to stochastic fluctuations. At the time when the polymer fibrils coalesce / nucleate into a chiral mesophase, such population bias can be amplified through conversion of the minority into the majority handedness, driven by intermolecular interactions depicted in Figure 6 and free energy minimization when forming a coherent mesophase with uniform handedness. This view is supported by the observed stochastic nature of chiral symmetry breaking – the twist-bent mesophases I, II and striped twist-bent mesophase can all adopt both handedness with certain probabilities. It remains a question, however, whether the experimentally observed biases towards one handedness in a chiral mesophase relates to asymmetries at the molecular scale, namely asymmetric dihedral angle distribution and asymmetric backbone stacking observed in MD simulations. While such molecular asymmetries do not lead to local symmetry breaking at the single polymer or single

nanofibril level, nuanced imbalances at the molecular level could be amplified during the multimolecular assembly process. Indeed, we observed that bias towards certain handedness increases with increasing volume fraction/concentration of the polymer in solution. This suggests that asymmetric intermolecular interactions may play an important role in chiral symmetry breaking”.

References

1. Park, K. S., Kwok, J. J., Dilmurat, R., Qu, G., Kafle, P., Luo, X. Y., Jung, S. H., Olivier, Y., Lee, J. K., Mei, J. G., Beljonne, D. & Diao, Y. Tuning conformation, assembly, and charge transport properties of conjugated polymers by printing flow. *Sci Adv* **5** (2019).
2. Classen, A., Chochos, C. L., Luer, L., Gregoriou, V. G., Wortmann, J., Osvet, A., Forberich, K., McCulloch, I., Heumuller, T. & Brabec, C. J. The role of exciton lifetime for charge generation in organic solar cells at negligible energy-level offsets. *Nat Energy* **5**, 711-719 (2020).
3. Wang, P., Jeon, I., Lin, Z., Peeks, M. D., Savagatrup, S., Kooi, S. E., Van Voorhis, T. & Swager, T. M. Insights into magneto-optics of helical conjugated polymers. *J Am Chem Soc* **140**, 6501-6508 (2018).
4. Davidson, E. C., Rosales, A. M., Patterson, A. L., Russ, B., Yu, B. H., Zuckermann, R. N. & Segalman, R. A. Impact of helical chain shape in sequence-defined polymers on polypeptoid block copolymer self-assembly. *Macromolecules* **51**, 2089-2098 (2018).
5. Li, M. M., Balawi, A. H., Leenaers, P. J., Ning, L., Heintges, G. H. L., Marszalek, T., Pisula, W., Wienk, M. M., Meskers, S. C. J., Yi, Y. P., Laquai, F. & Janssen, R. A. J. Impact of polymorphism on the optoelectronic properties of a low-bandgap semiconducting polymer. *Nat Commun* **10** (2019).
6. Widge, A. S., Matsuoka, Y. & Kurnikova, M. Development and initial testing of an empirical forcefield for simulation of poly(alkylthiophenes). *J Mol Graph Model* **27**, 34-44 (2008).
7. Michaels, W., Zhao, Y. & Qin, J. Atomistic modeling of pedot:Pss complexes ii: Force field parameterization. *Macromolecules* **54**, 5354-5365 (2021).
8. Richter, L. J., DeLongchamp, D. M. & Amassian, A. Morphology development in solution-processed functional organic blend films: An in situ viewpoint. *Chem Rev* **117**, 6332-6366 (2017).
9. Spano, F. C. The spectral signatures of frenkel polarons in h- and j-aggregates. *Accounts Chem Res* **43**, 429-439 (2010).
10. Barford, W. & Marcus, M. Perspective: Optical spectroscopy in pi-conjugated polymers and how it can be used to determine multiscale polymer structures. *J Chem Phys* **146** (2017).
11. Spano, F. C. & Silva, C. H- and j-aggregate behavior in polymeric semiconductors. *Annu Rev Phys Chem* **65**, 477-500 (2014).

REVIEWER COMMENTS

Reviewer #2 (Remarks to the Author):

The authors have clearly addressed the technical concerns highlighted by the three reviewers. In particular, the analysis of equal probability of handedness and of dihedral correlations in MD simulations have significantly improved the manuscript. As such, the conclusions are well supported by the provided data.

This paper is a significant contribution to the field, as it is a significant advance in our understanding of the evolution of liquid crystalline phases in rigid polymers, such as conjugated polymers. The work opens the door for many further studies of lyotropic liquid crystalline polymers, and in taking advantage of their properties for various applications in optoelectronics. As the authors mention, there is evidence in the literature that such complex phases are present in many conjugated polymers, but are currently unreported. Given the prominence of solution-processing in organic electronics, it is likely that such assemblies impact structure, properties and function of these materials, meaning that this paper will spawn many follow-on studies that characterize the emergence of helical and liquid crystalline phases (and their consequences in applications).

There is only one more point that I believe the authors need to address – they should cite and briefly discuss the work by Madsen on rigid rod polymers that show helical assembly (see for example *Macromolecules* 47, 2984–2992 (2014) and *Nature Communications* 10, 801 (2019)).

Reviewer #3 (Remarks to the Author):

The authors have substantially revised their manuscript, embracing suggestions that I and other reviewers made in the first round of review and providing substantial new simulation and experimental data to further support their points. The revised manuscript is significantly more robust in its narrative, particularly around the origins and nature of the chiral emergence.

I am satisfied that the manuscript should proceed to publication provided the authors consider the note below.

Although I appreciate the addition of charge generation measurements to illustrate electronic consequences to helical assembly, I would caution the authors from promoting the notion that helical assembly alone can promote charge generation; the prose in pp25-26 is too strongly phrased that way.

Almost any conformational ordering process in organic electronics, including vanilla orthorhombic crystallization, can have similar effects. The main effect is the decrease of energetic disorder; the exciting new result here is that we can add helical assembly to the list of ordering processes that deliver lower extents of energetic disorder.

Point-by-Point Response for Reviewers' Comments in "Chiral emergence in multistep hierarchical assembly of achiral conjugated polymers"

Reviewer comments in blue, corresponding response below in black. Changes made to the manuscript and supplementary information (SI) indicated at the end of each comment.

Reviewer 2 Comments

The authors have clearly addressed the technical concerns highlighted by the three reviewers. In particular, the analysis of equal probability of handedness and of dihedral correlations in MD simulations have significantly improved the manuscript. As such, the conclusions are well supported by the provided data.

This paper is a significant contribution to the field, as it is a significant advance in our understanding of the evolution of liquid crystalline phases in rigid polymers, such as conjugated polymers. The work opens the door for many further studies of lyotropic liquid crystalline polymers, and in taking advantage of their properties for various applications in optoelectronics. As the authors mention, there is evidence in the literature that such complex phases are present in many conjugated polymers, but are currently unreported. Given the prominence of solution-processing in organic electronics, it is likely that such assemblies impact structure, properties and function of these materials, meaning that this paper will spawn many follow-on studies that characterize the emergence of helical and liquid crystalline phases (and their consequences in applications).

There is only one more point that I believe the authors need to address – they should cite and briefly discuss the work by Madsen on rigid rod polymers that show helical assembly (see for example *Macromolecules* 47, 2984–2992 (2014) and *Nature Communications* 10, 801 (2019)).

Response: We greatly appreciate the positive comments on our work. We also thank the reviewer for providing these insightful references. We address this comment in the revised manuscript.

Changes made to the manuscript:

We included those works with the related discussion in the main manuscript (page 2).

Reviewer 3 Comments

The authors have substantially revised their manuscript, embracing suggestions that I and other reviewers made in the first round of review and providing substantial new simulation and experimental data to further support their points. The revised manuscript is significantly more robust in its narrative, particularly around the origins and nature of the chiral emergence.

I am satisfied that the manuscript should proceed to publication provided the authors consider the note below.

Although I appreciate the addition of charge generation measurements to illustrate electronic consequences to helical assembly, I would caution the authors from promoting the notion that helical assembly alone can promote charge generation; the prose in pp25-26 is too strongly phrased that way.

Almost any conformational ordering process in organic electronics, including vanilla orthorhombic crystallization, can have similar effects. The main effect is the decrease of energetic disorder; the exciting new result here is that we can add helical assembly to the list of ordering processes that deliver lower extents of energetic disorder.

Response: We greatly appreciate the positive comments on our work. We agree with the reviewer that our discussion seemed too strong to conclude in that way. We address this comment in the revised manuscript.

Changes made to the manuscript:

We revised the corresponding discussion in the main manuscript (page 19) to soften the claim and added the possible reason for the observed enhancement per reviewer's comments.